



# Technical note: Gas-phase nitrate radical generation via irradiation of aerated ceric ammonium nitrate mixtures

Andrew T. Lambe[1], Bin Bai[2], Masayuki Takeuchi[3], Nicole Orwat[4], Paul M. Zimmerman[4], Mitchell W. Alton[1], Nga L. Ng[2,3,5], Andrew Freedman[1], Megan S. Claflin[1], Drew R. Gentner[6,7], Douglas R. Worsnop[1], and Pengfei Liu[2]

[1]Aerodyne Research, Inc., Billerica, MA, USA
[2]School of Earth and Atmospheric Sciences, Georgia Institute of Technology, Atlanta, GA, USA
[3]School of Civil and Environmental Engineering, Georgia Institute of Technology, Atlanta, GA, USA
[4]Department of Chemistry, University of Michigan, Ann Arbor, MI, USA
[5]School of Chemical and Biomolecular Engineering, Georgia Institute of Technology, Atlanta, GA, USA
[6]Department of Chemical and Environmental Engineering, Yale University, New Haven, CT, USA
[7]School of the Environment, Yale University, New Haven, CT, USA

**Correspondence:** Andrew T. Lambe (lambe@aerodyne.com)

**Abstract.**

We present a novel photolytic source of gas-phase $NO_3$ suitable for use in atmospheric chemistry studies that has several advantages over traditional sources that utilize $NO_2 + O_3$ reactions and/or thermal dissociation of dinitrogen pentoxide ($N_2O_5$). The method generates $NO_3$ via irradiation of aerated aqueous solutions of ceric ammonium nitrate (($NH_4)_2Ce(NO_3)_6$,
"CAN") and nitric acid ($HNO_3$) or sodium nitrate ($NaNO_3$). We present experimental and model characterization of the $NO_3$ formation potential of irradiated CAN/$HNO_3$ and CAN/$NaNO_3$ mixtures containing [CAN] = $10^{-3}$ to 1.0 M, [$HNO_3$] = 1.0 to 6.0 M, [$NaNO_3$] = 1.0 to 4.8 M, photon fluxes ($I$) ranging from $6.9 \times 10^{14}$ to $1.0 \times 10^{16}$ photons cm$^{-2}$ s$^{-1}$, and irradiation wavelengths ranging from 254 to 421 nm. $NO_3$ mixing ratios ranging from parts per billion to parts per million by volume were achieved using this method. At the CAN solubility limit, maximum [$NO_3$] was achieved using [$HNO_3$] $\approx$ 3.0 to 6.0 M
and UVA radiation ($\lambda_{max}$ = 369 nm) in CAN/$HNO_3$ mixtures or [$NaNO_3$] $\geq$ 1.0 M and UVC radiation ($\lambda_{max}$ = 254 nm) in CAN/$NaNO_3$ mixtures. Other reactive nitrogen ($NO_2$, $N_2O_4$, $N_2O_5$, $N_2O_6$, $HNO_2$, $HNO_3$, $HNO_4$) and reactive oxygen ($HO_2$, $H_2O_2$) species obtained from the irradiation of ceric nitrate mixtures were measured using a $NO_x$ analyzer and an iodide adduct high-resolution time-of-flight chemical ionization mass spectrometer (HR-ToF-CIMS). To assess the applicability of the method for studies of $NO_3$-initiated oxidative aging processes, we generated and measured the chemical composition of
oxygenated volatile organic compounds and secondary organic aerosols from the $\beta$-pinene + $NO_3$ reaction using a Filter Inlet for Gases and Aerosols (FIGAERO) coupled to the HR-ToF-CIMS.

## 1  Introduction

The importance of $NO_3$ as a nighttime atmospheric oxidant is well established (Wayne et al., 1991; Brown and Stutz, 2012; Ng et al., 2017; Wang et al., 2023). $NO_3$ is generated via the reaction $NO_2 + O_3 \rightarrow NO_3 + O_2$, followed by achievement of



temperature-dependent equilibrium between $NO_3$, $NO_2$, and dinitrogen pentoxide ($N_2O_5$). $N_2O_5$ also hydrolyzes efficiently to $HNO_3$ on aqueous surfaces (Brown et al., 2004). Thus, any investigation of the influence of $NO_3$ chemistry in a specific source region necessarily must account for the local temperature, humidity, and particle surface area along with other factors. Despite these complications, for decades, laboratory studies investigating gas-phase $NO_3$ chemistry have utilized the same $NO_2 + O_3$ reactions and/or $N_2O_5$ thermal decomposition to produce $NO_3$ as occurs in the atmosphere, and accommodated the

inherent limitations associated with $N_2O_5$; namely, that it must be stored under cold and dry conditions until use. Few viable alternative methods for the generation of gas-phase $NO_3$ have been identified. Reactions between fluorine atoms and nitric acid ($F + HNO_3 \rightarrow HF + NO_3$), or chlorine atoms and chlorine nitrate ($Cl + ClNO_3 \rightarrow Cl_2 + NO_3$) require handling and/or synthesizing hazardous halogen-containing compounds (Burrows et al., 1985; Bedjanian, 2019). F and Cl can also compete with $NO_3$ for the oxidation of target analytes, as can $O_3$ if its reaction with $NO_2$ is used as the $NO_3$ source.

In the 1960s and 1970s, following earlier research into the properties of ceric solutions (Meyer and Jacoby, 1901; Wylie, 1951; Hinsvark and Stone, 1956; Blaustein and Gryder, 1957), Thomas Martin and coworkers discovered that irradiating solutions containing ceric ammonium nitrate (CAN, $(NH_4)_2Ce(NO_3)_6$) generates aqueous $NO_3$ (Henshall, 1963; Martin et al., 1963, 1964; Glass and Martin, 1970; Martin and Glass, 1970; Martin and Stevens, 1978). In $\gtrsim$ 6M nitric acid ($HNO_3$), CAN is thought to dissociate primarily into $NH_4^+$ cations and hexanitratocerate ($Ce(NO_3)_6^{2-}$) anions (Henshall, 1963). The

$Ce(NO_3)_6^{2-}$ is subsequently reduced to $Ce(NO_3)_5^{2-}$ upon irradiation by ultraviolet light, and $NO_3$ is generated as a primary photolysis product. A similar process occurs in other solvents, although the ensuing ceric composition in solution is complex and influenced by several factors. For example, in glacial acetic acid ($CH_3COOH$), CAN dissociates into primarily $Ce(NO_3)_4$ (Henshall, 1963). Additionally, ceric ions containing complexed hydroxyl (OH) or $H_2O$, $CH_3COOH$, or acetonitrile ($CH_3CN$) molecules are formed in aqueous, acetic acid, or $CH_3CN$ media, respectively (Henshall, 1963; Glebov et al., 2021). Higher

solution acidity and/or CAN concentration appears to promote the formation of $Ce(NO_3)_6^{2-}$ (Wylie, 1951) and ceric nitrate dimers (Blaustein and Gryder, 1957; Demars et al., 2015). The following generalized mechanism was proposed by Glass and Martin (1970) to describe ceric nitrate photochemistry:

$$Ce^{(IV)} + h\nu \quad \rightarrow \quad Ce^{(III)} + NO_3 \tag{R1}$$

$$Ce^{(III)} + NO_3 \quad \rightarrow \quad Ce^{(IV)} \tag{R2}$$

$$NO_3 + NO_3 \quad \rightarrow \quad N_2O_6 \tag{R3}$$

$$N_2O_6 + 2Ce^{(IV)} \quad \rightarrow \quad 2NO_2 + O_2 + 2Ce^{(III)} \tag{R4}$$

$$NO_3 + NO_2 + H_2O \quad \rightarrow \quad 2HNO_3 \tag{R5}$$

where $Ce^{(IV)}$ represents ceric nitrates as diverse as $Ce(NO_3)_4$, $Ce(NO_3)_6^{2-}$, $(NO_3)_5CeOCe(NO_3)_5^{4-}$, and $(H_2O)_3(NO_3)_3CeOCe(NO_3)_3(H_2O)_3$ that are potentially formed in solution (Henshall, 1963; Blaustein and Gryder, 1957;

Demars et al., 2015). Similarly, $Ce^{(III)}$ represents cerous nitrates such as $Ce(NO_3)_3$ and $Ce(NO_3)_5^{2-}$. The rate of Reaction R2 is [$HNO_3$]-dependent (Martin and Glass, 1970), and the dinitrogen hexaoxide ($N_2O_6$) intermediate was proposed on the basis of supporting observations without direct measurements (Glass and Martin, 1970).





CAN is used routinely as an oxidizing agent in organic synthesis due to its widespread availability and low cost, high oxidative potential, and low toxicity (Nair and Deepthi, 2007). However, its usage in atmospheric chemistry to date is limited

to studies of $NO_3$-initiated oxidative aging processes in solution, e.g. Alexander (2004). Given the potential simplicity of irradiating $Ce^{(IV)}$ mixtures relative to synthesizing and storing $N_2O_5$ under cold and dry conditions or reacting $NO_2 + O_3$ under carefully controlled conditions, $Ce^{(IV)}$ irradiation could in principle enable more widespread studies of $NO_3$ oxidation chemistry, which is understudied compared to OH chemistry (Ng et al., 2017). Here, for the first time, we investigated the use of $Ce^{(IV)}$ irradiation as a source of gas-phase $NO_3$. First, we designed a photoreactor that generates gas-phase $NO_3$ from

irradiated CAN/$HNO_3$ and CAN/$NaNO_3$ mixtures. Second, we characterized $NO_3$ concentrations achieved over a range of reactor operating conditions and mixture composition. Third, we characterized gas-phase reactive nitrogen and reactive oxygen species generated following $Ce^{(IV)}$ irradiation. Fourth, we demonstrated application of the method to generate and characterize OVOCs and SOA from the $\beta$-pinene + $NO_3$ reaction.

## 2  Methods

### 2.1  Photoreactor design and operation

Figure 1 shows a schematic of the experimental setup used in this study. A zero air carrier gas flow of 0.5 L min$^{-1}$ was bubbled through a gas dispersion line consisting of 6.35 mm OD x 4.8 mm ID FEP tubing into approximately 10 mL of aqueous CAN/$HNO_3$ or CAN/$NaNO_3$ mixtures placed at the bottom of a 12.7 mm OD x 11.1 cm ID FEP tube. The FEP tube was surrounded by low-pressure mercury fluorescent lamps installed vertically in a custom enclosure. These lamps had a 35.6 cm

illuminated length. At these operating conditions, the calculated gas transit time in the illuminated portion of the reactor was approximately 3 s. After exiting the photoreactor, the carrier gas flow was passed through a filter holder (Savillex, 401-21-47-10-21-2) containing a 47 mm PTFE membrane filter (Pall Gelman, R2PJ047) to transmit $NO_3$ (Wagner et al., 2011) while removing stray droplets from the sample flow. At the end of each experiment, the lamps were turned off, the gas dispersion line was removed from the top of the reactor, and FEP tubing and filter holder were flushed with distilled $H_2O$ to remove residual

$Ce^{(III)}$ precipitate. Initial studies were conducted using a Cavity Attenuated Phase Shift (CAPS) $NO_2$ monitor operating at $\lambda$ = 405 nm (Kebabian et al., 2005) and a second retrofitted CAPS monitor operating at $\lambda$ = 630 nm which established that $NO_2$ and $NO_3$ were produced from irradiated $Ce^{(IV)}$. Subsequent studies described in the next section used a 2B Technologies Model 405 analyzer to measure NO and $NO_2$ (Birks et al., 2018).

Depending on the specific experiment, lamps with peak emission output centered at $\lambda$ = 254, 313, 369, or 421 nm, re-

spectively (GPH436TL/4P, Light Sources, Inc.; F436T5/NBUVB/4P-313, F436T5/BLC/4P-369, F436T5/SDI/4P-421, LCD Lighting, Inc.) were used. Emission spectra from the manufacturer are shown in Figure S1. A fluorescent dimming ballast (IZT-2S28-D, Advance Transformer Co.) was used to regulate current applied to the lamps. To quantify the photon flux $I$ in the photoreactor for studies that used $\lambda$ = 254, 313, or 369 nm radiation, we measured the rate of externally added $O_3$ ($\lambda$ = 254 nm) or $NO_2$ photolysis ($\lambda$ = 313 or 369 nm) as a function of lamp voltage under dry conditions (RH < 5%). $NO_2$ photolysis

measurements were conducted in the absence of oxygen to avoid $O_3$ formation. $I$-values were then calculated using methods



described in Lambe et al. (2019); maximum $I_{254} = 1.0 \times 10^{16}$ photons cm$^{-2}$ s$^{-1}$, $I_{313} = 6.0 \times 10^{15}$ photons cm$^{-2}$ s$^{-1}$, and $I_{369} = 7.0 \times 10^{15}$ photons cm$^{-2}$ s$^{-1}$ were obtained.

## 2.2 Characterization studies

In one set of experiments, the 0.5 L min$^{-1}$ photoreactor effluent was mixed with a 6.5 L min$^{-1}$ zero air carrier gas and injected into a dark Potential Aerosol Mass oxidation flow reactor (OFR; Aerodyne Research, Inc.), which is a horizontal 13 L Teflon-coated aluminum cylindrical chamber operated in continuous flow mode. Approximately 6.5 L min$^{-1}$ of sample flow was pulled from the reactor, resulting in a calculated mean residence time in the OFR ($\tau_{OFR}$) of approximately 120 s. To constrain NO$_3$ mixing ratios, a mixture of 10 VOC tracers with NO$_3$ reaction rate coefficients ($k_{NO_3}$) ranging from $3.01 \times 10^{-19}$ to $2.69 \times 10^{-11}$ cm$^3$ molecules$^{-1}$ s$^{-1}$ at $T = 298$ K (Table S2) was injected through a 10.2 cm length of 0.0152 cm ID Teflon tubing at a liquid flow rate of 0.94 $\mu$L hr$^{-1}$ using a syringe pump. The tracer mixture was then evaporated into a 1 L min$^{-1}$ zero air carrier gas prior to injection into the OFR. The total external NO$_3$ reactivity (NO$_3$R$_{ext}$), which is the summed product of each tracer mixing ratio and its $k_{NO_3}$, was approximately 5 s$^{-1}$. VOCs with proton affinities greater than that of H$_2$O were chosen to enable their measurement with a Tofwerk/Aerodyne Vocus proton transfer-reaction time-of-flight mass spectrometer (hereafter referred to as "Vocus PTR") operated using H$_3$O$^+$ reagent ion chemistry (Krechmer et al., 2018) and $\sim$ 8000 (Th/Th) resolving power. NO$_3$ mixing ratios were calculated from the measured decrease in VOC mixing ratios using the Vocus PTR. Here, we assumed that the total concentration of reacted VOCs was equal to the concentration of NO$_3$ injected into the OFR; because NO$_3$ may additionally react with organic peroxy radicals (RO$_2$) generated from VOC + NO$_3$ reactions as well as OVOCs, these calculated NO$_3$ concentrations represent lower limits. A subset of OVOCs generated from VOC + NO$_3$ reactions that had proton affinities greater than that of H$_2$O were also detected with the Vocus PTR.

In a separate set of experiments, the photoreactor effluent was diluted into 4 L min$^{-1}$ zero air carrier gas and sampled with an Aerodyne iodide-adduct high-resolution time-of-flight chemical ionization mass spectrometer (HR-ToF-CIMS; hereafter referred to as "CIMS"; Bertram et al. (2011)) and the NO$_x$ analyzer. The CIMS was operated at a $\sim$ 4000 (Th/Th) resolving power. Iodide-adduct reagent ion chemistry was used due to its high sensitivity and selectivity towards nitrogen oxides and multifunctional organic nitrates (Lee et al., 2014). To demonstrate application of the method to study NO$_3$-initiated oxidative aging processes, the chemical composition of $\beta$-pinene + NO$_3$ gas-and-condensed-phase oxidation products was measured with a Filter Inlet for Gases and Aerosols (FIGAERO) coupled to the CIMS (Lopez-Hilfiker et al., 2013). Gas sampling and simultaneous particle collection was performed for 1 min intervals, followed by thermal desorption of the particle sample from a PTFE filter membrane (15 min ramp from room temperature to 200°C, 10 min holding time, 8 min cooldown to room temperature).

## 2.3 Photochemical model

To supplement our measurements, and to characterize aqueous phase concentrations of species produced in the photoreactor that were not measured, we developed a photochemical box model that was implemented in the KinSim chemical kinetic solver (Peng and Jimenez, 2019). The KinSim mechanism shown in Table S1 contains 79 reactions to model concentrations of Ce$^{(IV)}$,





$Ce^{(III)}$, NO, $NO_2$, $NO_3$, $N_2O_3$, $N_2O_4$, $N_2O_5$, $HNO_2$, $HNO_3$, $HNO_4$, H, O, OH, $HO_2$, and $H_2O_2$. We assumed that $HNO_3$
that was present in solution prior to irradiation completely dissociated into $H^+$ and $NO_3^-$. When possible, we used condensed-
phase rate coefficients in the mechanism. For reactions that we assumed occurred but did not have published condensed-phase
rate coefficients (e.g. $NO_3 + OH \rightarrow NO_2 + HO_2$) we used published gas-phase rate coefficients instead with no modifications
aside from unit conversion. Gas-phase wall loss rates of $NO_x$, $NO_y$, and $HO_x$ species were not explicitly considered in the
mechanism. UV/Vis extinction cross sections ($\sigma_{ext}$) of CAN/$HNO_3$ and CAN/$NaNO_3$ mixtures were separately obtained
between $\lambda = 200$ and 600 nm using an Agilent Cary 5000 UV/Vis/NIR spectrophotometer. Because of the high absorptivity
and concentrations of the mixtures, samples were prepared in a 0.01 mm short-path-length cuvette (20/C-Q-0.01, Starna) to
minimize saturation of the photodetector relative to a cuvette with a standard 10 mm path length. Even with the cuvette that was
used, CAN dilution was necessary in some cases in order to obtain $\sigma_{ext}$ without photodetector saturation at shorter wavelengths.
Spectra were obtained as a function of [CAN] (0.047 to 0.526 M), [$HNO_3$] (0 to 6.0 M), and [$NaNO_3$] (0 to 4.0 M) to cover
the approximate range of mixture compositions that were characterized in Section 2.2. The $\sigma_{ext}$-values of the mixtured were
then calculated using the Beer-Lambert law and applied in the KinSim mechanism. Model outputs were obtained over a total
experimental time of 14400 s at 1 s intervals.

## 3    Results and Discussion

The maximum $NO_3$ quantum yield ($\phi_{NO_3}$) of UVA-irradiated CAN/$HNO_3$ mixtures is obtained at 6.0 M $HNO_3$ (Martin
and Stevens, 1978); thus, this mixture composition served as the basis from which additional characterization studies were
conducted. We found that 0.5 M CAN was the approximate solubility limit in 6.0 M $HNO_3$ at 25°C. Because 1.1 M CAN
is the solubility limit in $H_2O$ and CAN is almost nearly in $HNO_3$ (Martin and Glass, 1970), 0.7 M CAN is the estimated
solubility limit in 6.0 M $HNO_3$ in the absence of changes in ceric nitrate composition in solution. Thus, the reduction in CAN
solubility (0.7 M $\rightarrow$ 0.5 M) observed in our studies was presumably associated with significant conversion of CAN to dimeric
ceric nitrates in 6.0 M $HNO_3$ (Blaustein and Gryder, 1957; Demars et al., 2015).

### 3.1    $NO_3$ characterization studies

Figure 2a shows time series of thiophene ($C_4H_4S$), 2,3-dihydrobenzofuran ($C_8H_8O$), cis-3-hexenyl acetate ($C_8H_{14}O_2$), iso-
prene ($C_5H_8$), dimethyl sulfide ($C_2H_6S$), 2,5-dimethylthiophene ($C_6H_8S$), $\alpha$-pinene ($C_{10}H_{16}$), and guaiacol ($C_7H_8O_2$) con-
centrations following injection into the OFR and exposure to $NO_3$ generated in the photoreactor from irradiation of a mixture
of 0.5 M CAN and 6.0 M $HNO_3$ at $I_{369} = 7 \times 10^{15}$ photons $cm^{-2}$ $s^{-1}$. Here, concentrations of each VOC were first normalized
to the acetonitrile concentration to correct for changes in the syringe pump output over time and then normalized to the VOC
concentration prior to $NO_3$ exposure. Aside from $C_6H_8S$, whose relative decay was less pronounced than expected (Table
S2), and butanal ($C_4H_8O$, not shown), whose signal decreased by approximately 30% and did not recover for reasons that
are unclear, the oxidative loss of each tracer increased with increasing $k_{NO_3}$. Maximum tracer consumption was observed at
the beginning of the experiment due to maximum $NO_3$ production from $Ce^{(IV)}$ irradiation. As the experiment progressed and



$Ce^{(IV)}$ was reduced to $Ce^{(III)}$, the $NO_3$ concentration and corresponding VOC oxidative loss decreased. Compared to the other VOCs, the initial increase in $C_{10}H_{16}$ and $C_7H_8O_2$ concentrations over the first 2 hours was delayed because of their higher $k_{NO_3}$ values that resulted in >95% consumption and lower sensitivity to changes in $[NO_3]$ in the initial stage of the experiment. To confirm that VOC degradation shown in Fig. 2a was due to reaction with $NO_3$, Figure S2 shows the relative

$NO_3$ rate coefficients obtained from the decay of $C_4H_4S$, $C_8H_8O$, and $C_8H_{14}O_2$ measured with the Vocus PTR. We measured relative rate coefficients of 3.59 between $C_8H_8O$ and $C_4H_4S$ and 6.92 between $C_8H_{14}O_2$ and $C_4H_4S$, which are in agreement with relative rate coefficient values of $3.44\pm1.20$ and $7.68\pm2.84$ calculated from their absolute $NO_3$ rate coefficients (Atkinson, 1991; D'Anna et al., 2001). Time series of ions corresponding to nitrothiophene ($C_4H_3NO_2S$), $C_5H_7NO_{4-6}$ and $C_{10}H_{15}NO_{5,6}$ organic nitrates, and nitroguaiacol ($C_7H_7NO_4$), which are known $NO_3$ oxidation products of $C_4H_4S$, $C_5H_8$,

$C_{10}H_{16}$, and $C_7H_8O_2$ (Atkinson et al., 1990; Jenkin et al., 2003; Saunders et al., 2003; Cabañas et al., 2005), along with $C_8H_{5,7}NO_{4-6}$ and $C_8H_{13}NO_{5-6}$ ions that may be associated with $NO_3$ oxidation products of $C_8H_8O$ and $C_8H_{14}O_2$, respectively, were anticorrelated with those of their respective VOC precursors (Figure S3). Tracer decay experiments similar to the one shown in Figure S2 were used to obtain results that are discussed in more detail in Sections 3.2, 3.3, and 3.4.

## 3.2  Effect of irradiation wavelength

Figure 3a shows normalized $[NO_3]$ values obtained following irradiation of mixtures containing CAN and 6.0 M $HNO_3$ or 4.8 M $NaNO_3$ as a function of irradiation wavelength. In CAN/$HNO_3$ mixtures, $[NO_3]$ was a factor of 2.4-3.5 higher following irradiation at $\lambda = 369$ compared to the other wavelengths. On the other hand, $[NO_3]$ decreased with increasing irradiation wavelength following irradiation of CAN/$NaNO_3$ mixtures; at $\lambda = 254$ nm, $[NO_3]$ was a factor of 3.2-42 times higher than at the other irradiation wavelengths that were used. These differences in $[NO_3]$ were larger than the differences in calibrated

$I$-values at the maximum ouptut of each lamp type ($\pm40\%$; Sect. 2.1). Different $Ce^{(IV)}$ in CAN/$HNO_3$ and CAN/$NaNO_3$ mixtures may have influenced these trends, as suggested by their UV/Vis spectra (Fig. 3b). The $\sigma_{ext}$ curves of CAN/$HNO_3$ mixtures were generally larger, broader, and red-shifted relative to those of CAN/$NaNO_3$ mixtures, with the extent of red-shifting increasing with increasing $[HNO_3]$, possibly due to higher yields of $Ce(NO_3)_6^{2-}$ and/or ceric nitrate dimers (Blaustein and Gryder, 1957; Henshall, 1963; Demars et al., 2015). For $\lambda > 250$ nm, CAN/$HNO_3$ mixtures had $\sigma_{ext,max}$ values between

$\lambda = 306$ - 311 nm, whereas CAN/$NaNO_3$ solutions had $\sigma_{ext,max}$ values at $\lambda = 296$ nm. However, if $[NO_3]$ was simply proportional to $\sigma_{ext}$, irradiation of CAN/$HNO_3$ mixtures at $\lambda = 313$ nm should have produced the highest $[NO_3]$; this was not the case. Instead, model calculations suggest that higher $[NO_2]$ obtained from significantly faster photolysis of $HNO_3$ at $\lambda = 254$ and 313 nm relative to $\lambda > 350$ nm suppressed $NO_3$ downstream of the photoreactor when shorter irradiation wavelengths were used (Sander et al. (2011), Table S1). At a photon flux of $10^{16}$ photons cm$^{-2}$ s$^{-1}$, model-calculated $[NO_3]$

values were within $\pm$ 13% of each other for irradiation wavelengths ranging from $\lambda = 254$ to 369 nm. However, higher $[NO_2]$ values obtained following $Ce^{(IV)}$ irradiation at $\lambda = 254$ and 313 nm suppressed $NO_3$ by >96% relative to the $\lambda = 369$ nm case during 120 s of simulated $NO_2 + NO_3$ reactions in the OFR. Thus, although the measured $NO_3$ suppression at these other irradiation wavelengths was less substantial than the model output, the measurement and model trends, along with achievement



of maximum [NO$_3$] following $\lambda$ = 254 nm irradiation of CAN/NaNO$_3$ mixtures that had lower [HNO$_3$], qualitatively support

this explanation for the wavelength-dependent NO$_3$ yields observed in CAN/HNO$_3$ mixtures.

### 3.3 Effect of mixture composition

To characterize the influence of individual reagents on NO$_3$ formation, tracer decay experiments similar to the measurements shown in Figure 2 were repeated as a function of [CAN], [HNO$_3$], and [NaNO$_3$]. Figure 4a shows [NO$_3$] obtained from irradiated 6.0 M HNO$_3$ solutions containing 0.001 to 0.5 M CAN ($I_{369}$ = 7×10$^{15}$ photons cm$^{-2}$ s$^{-1}$), and irradiated 1.0

M NaNO$_3$ solutions containing 0.5 to 1.0 M CAN ($I_{254}$ = 1×10$^{16}$ photons cm$^{-2}$ s$^{-1}$). Results were normalized to [NO$_3$] achieved with solutions containing 0.5 M CAN and 6.0 M HNO$_3$. Control experiments conducted with irradiated 6.0 M HNO$_3$ or 1.0 M NaNO$_3$ solutions at $I_{254}$ = 1×10$^{16}$ photons cm$^{-2}$ s$^{-1}$ in the absence of CAN suggest that a fraction of the NO$_3$ obtained in CAN mixtures was generated via the reactions HNO$_3$ + h$\nu$ → OH + NO$_2$ and HNO$_3$ + OH → NO$_3$ + H$_2$O. The remaining NO$_3$ was clearly obtained from CAN irradiation because [NO$_3$] increased with increasing [CAN], as expected from

Reaction R1. Overall, [NO$_3$] increased by approximately a factor of 3 as [CAN] was increased from 0.001 to 0.5 M in 6.0 M HNO$_3$.

Figure 4b shows [NO$_3$] obtained in irradiated solutions containing 0.5 M CAN as a function of [HNO$_3$] ranging from 1.0 to 6.0 M or [NaNO$_3$] ranging from 1.0 to 4.8 M at the same $I_{369}$ and $I_{254}$ values used to obtain results shown in Fig. 4a. Irradiated CAN solutions containing 3.0 M and 6.0 M HNO$_3$ generated the same [NO$_3$] concentrations within measurement

uncertainties, presumably because the NO$_3$ quantum yield ($\phi_{NO_3}$) ranged from 0.92-1.00 over this range of acidity (Martin and Stevens, 1978; Wine et al., 1988). [NO$_3$] decreased by a factor of 2 as [HNO$_3$] was decreased from 3.0 M to 1.0 M, consistent with a reduction in $\phi_{NO_3}$ from 0.92 to 0.46 (Martin and Stevens, 1978). On the other hand, in irradiated CAN/NaNO$_3$ mixtures with uncharacterized $\phi_{NO_3}$, [NO$_3$] was constant within measurement uncertainties between 1.0 and 4.8 M NaNO$_3$.

Other mixture components that were tested or considered included substitution of CH$_3$CN in place of H$_2$O and HNO$_3$,

ammonium nitrate (NH$_4$NO$_3$) instead of NaNO$_3$, ceric potassium nitrate (K$_2$Ce(NO$_3$)$_6$) instead of CAN, and addition of sodium persulfate (Na$_2$S$_2$O$_8$) to generate additional NO$_3$ via S$_2$O$_8^{2-}$ + h$\nu$ → 2SO$_4^-$ followed by SO$_4^-$ + NO$_3^-$ → NO$_3$ + SO$_4^{2-}$ (Gaillard de Sémainville et al., 2007). CAN/CH$_3$CN mixtures are commonly used in organic synthesis applications, perhaps even more so than CAN/HNO$_3$ (Baciocchi et al., 1988; Choidini et al., 1993; Alexander, 2004). In limited testing, CAN/CH$_3$CN appeared to generate significantly less NO$_3$ than CAN/HNO$_3$ or CAN/NaNO$_3$, possibly due to lower $\phi_{NO_3}$

of irradiated Ce$^{(IV)}$-CH$_3$CN complexes (Glebov et al., 2021) and/or suppression of NO$_3$ due to its reaction with CH$_3$CN in solution. K$_2$Ce(NO$_3$)$_6$ is less widely available and less water-soluble than CAN and so was not considered further. Irradiation of CAN/NH$_4$NO$_3$ and CAN/NaNO$_3$ mixtures generated similar [NO$_3$], but we prefer NaNO$_3$ due to its lower volatilty. Finally, ternary mixtures containing 0.5 M CAN + 2.0 M NaNO$_3$ + 0.5 M Na$_2$S$_2$O$_8$ irradiated at $\lambda$ = 254 nm generated negligible additional NO$_3$ compared to binary CAN/NaNO$_3$ mixtures.



### 3.4 Effect of photon flux

Figure 5 shows normalized $[NO_3]$ values obtained from UVA-light-irradiated mixtures of 0.5 M CAN & 6.0 M $HNO_3$ and UVC-light-irradiated mixtures of 0.5 M CAN & 1.0 M $NaNO_3$ as a function of photon flux ranging from $6.9 \times 10^{14}$ to $7.5 \times 10^{15}$ photons $cm^{-2}$ $s^{-1}$. Results were normalized to $[NO_3]$ achieved with 0.5 M CAN, 6.0 M $HNO_3$ and $I_{369} = 6.8 \times 10^{15}$ photons $cm^{-2}$ $s^{-1}$. Symbols are colored by the $NO_3$ lifetime ($\tau_{NO_3}$), defined here as the time it took for $[NO_3]$ to experience one e-fold

decay relative to the maximum $[NO_3]$ that was measured. Figure 5 shows that $[NO_3]$ increased with increasing photon flux, consistent with the fact that it is a primary photolysis product, along with a concurrent decrease in $\tau_{NO_3}$ due to faster reduction of $Ce^{(IV)}$ to $Ce^{(III)}$. For the CAN/$HNO_3$ system, $[NO_3]$ increased by a factor of 1.5 as $I_{369}$ was increased from $6.9 \times 10^{14}$ to $6.8 \times 10^{15}$ photons $cm^{-2}$ $s^{-1}$, in agreement with the model-calculated increase in $[NO_3]$ within measurement uncertainty. $\tau_{NO_3}$ decreased from 9 to 5 hr. For the CAN/$NaNO_3$ system, $[NO_3]$ increased by a factor of 1.9 as $I_{254}$ was increased from

$1.0 \times 10^{15}$ to $7.5 \times 10^{15}$ photons $cm^{-2}$ $s^{-1}$, and $\tau_{NO_3}$ decreased from 10 to 3 hr.

To examine concentrations of $NO_3$ and a subset of additional gas-phase photolysis products obtained over a wider range of conditions, Figure 6 plots model-calculated $[NO_3]$, $NO_2$:$NO_3$, $HO_2$:$NO_3$, and $N_2O_5$:$NO_3$ values as a function of photon flux ranging from $1 \times 10^{14}$ to $1 \times 10^{17}$ photons $cm^{-2}$ $s^{-1}$ following $\lambda$ = 254, 313, 369 and 421 nm irradiation of a mixture of 0.5 M CAN and 6.0 M $HNO_3$. When considering only the primary photochemical process (Reactions R1-R5), maximum $[NO_3]$

values within ± 10% of each other were achieved at photon fluxes ranging from $5 \times 10^{15}$ ($\lambda$ = 313 nm) to $4 \times 10^{16}$ photons $cm^{-2}$ $s^{-1}$ ($\lambda$ = 421 nm). $[NO_3]$ values decreased at higher $I$-values due to conversion of $NO_3$ to $NO_2$ via photolysis. As shown in Fig. 6b, significant additional $NO_2$ production was obtained via $HNO_3$ photolysis at shorter irradiation wavelengths above $I \approx 10^{15}$ photons $cm^{-2}$ $s^{-1}$, resulting in $NO_2$:$NO_3$ > 10 ($\lambda$ = 254 nm) and 1 ($\lambda$ = 313 nm). Given additional reaction time downstream of the photoreactor, high $NO_2$ may suppress $NO_3$ (Sect. 3.2) and increase $N_2O_5$:$NO_3$ beyond the range of

values shown in Fig. 6c. We also calculated OH:$NO_3$ and $HO_2$:$NO_3$ following irradiation of CAN/$HNO_3$ mixtures over the range of conditions shown in Figure 6. Aqueous OH:$NO_3 \approx 0.1$ and did not change significantly as a function of photon flux or irradiation wavelength, and aqueous $HO_2$:$NO_3$ values ranged from 0.05 ($\lambda$ = 254 nm) to 0.25 ($\lambda \geq$ 369 nm). While OH influenced aqueous-phase chemistry inside the photoreactor via formation of reactive oxygen species (Sect. 3.5), OH probably did not influence downstream gas-phase chemistry due to significant wall losses inside the photoreactor: assuming a lower-limit

OH wall loss rate coefficient of 5 $s^{-1}$ (Schwab et al., 1989), the estimated OH penetration efficiency through the reactor was less than $10^{-6}$. Similarly, in studies involving the generation of $RO_2$ via VOC + $NO_3$ reactions, $HO_2$ is unlikely to significantly influence $RO_2$ fate because $RO_2$ + $HO_2$ reactions are several times slower than those of $RO_2$ + $NO_3$ reactions (Orlando and Tyndall, 2012).

### 3.5 Characterization of reactive nitrogen and reactive oxygen photolysis products

Figure S4 shows time series of $I^-$, $IH_2O^-$, $IO^-$, $IO_2^-$, $NO_2^-$, $NO_3^-$, $IHNO_3^-$, and $HNO_3NO_3^-$ obtained with the CIMS following irradiation of a mixture of 0.5 M CAN and 1.0 M $NaNO_3$ ($I_{254} \approx 10^{16}$ photons $cm^{-2}$ $s^{-1}$). Signals of $I^-$, $IH_2O^-$ and $IHNO_3^-$ decreased following irradiation of the CAN/$NaNO_3$ mixture, whereas $IO^-$, $IO_2^-$, $NO_2^-$, $NO_3^-$, and $HNO_3NO_3^-$



signals increased. One potential source of $IO_x^-$ is $I^-$ + $O_3$ reactions in the CIMS ion-molecule reactor (IMR); if this reaction

was the sole source of $IO_x^-$ here, we estimate an upper limit $O_3$ mixing ratio of approximately 15 ppbv present in the IMR

(Dörich et al., 2021). $NO_2^-$ is generated following the reaction of $I^-$ and/or $IO_x^-$ with $HNO_2$ (Abida and Osthoff, 2011), and

$NO_3^-$ is generated from the reaction of $I^-$ and/or $IO_x^-$ with multiple nitrogen oxides, including $NO_3$, $HNO_3$, $HNO_4$, and $N_2O_5$

(Huey et al., 1995; Veres et al., 2015; Dörich et al., 2021), all of which are generated following $Ce^{(IV)}$ irradiation as discussed

later in this section. Figure 7 shows time series of reactive nitrogen and reactive oxygen species detected following irradiation

of the same mixture of 0.5 M CAN and 1.0 M $NaNO_3$, shown here because the signal-to-noise in CIMS measurements of

irradiated $CAN/NaNO_3$ mixtures was generally better than in measurements of irradiated $CAN/HNO_3$ mixtures due to reagent

ion depletion by $HNO_3$. The $NO_2$ mixing ratio reached a maximum value of 26 ppbv shortly after the lights were turned on

(Fig. 7a), suggesting an initial $NO_2:NO_3 \approx 0.37$ (Fig. 4) that was similar to modeled $NO_2:NO_3 = 0.33$ obtained from irradiated

$CAN/HNO_3$ (Fig. 6). Multiple reactions may generate $NO_2$, including Reaction R3, $HNO_3$ and/or $NO_3$ photolysis, and other

reactions listed in Table S1. While $NO_2$ and/or $HNO_2$ photolysis generated NO, its concentration was negligible in these

experiments.

Figure 7b shows time series of $IN_2O_5^-$ and $IN_2O_6^-$ signals measured with the CIMS. Figure S4 additionally shows a time

series of $IN_2O_7^-$, and Figures S5, S6, and S7 shows high-resolution CIMS spectra at m/Q = 235, 251, and 267. $IN_2O_5^-$ was

formed from $NO_2$ + $NO_3 \rightarrow N_2O_5$ reactions in the photoreactor and $N_2O_5$ + $I^- \rightarrow IN_2O_5^-$ reactions in the CIMS IMR. As

expected, $IN_2O_5^-$ followed a similar profile as $NO_2$ and $NO_3$. Given $IN_2O_7^-:IN_2O_5^- \approx 10^{-3}$ coupled with similar $IN_2O_5^-$

and $IN_2O_7^-$ temporal profiles (Fig. S4), we hypothesize that $N_2O_5$ + $IO_x^-$ reactions in the IMR were the primary source of

$IN_2O_7^-$. $IN_2O_6^-$ was either generated from $NO_3$ + $NO_3 \rightarrow N_2O_6$ reactions in the photoreactor (Glass and Martin, 1970)

followed by $N_2O_6$ + $I^- \rightarrow IN_2O_6^-$ reactions in the IMR, or from the following series of reactions in the IMR: $HNO_3$ + $IO^-$

$\rightarrow NO_3^-$ + HOI, HOI + $NO_3^- \rightarrow INO_3$ + $OH^-$, and $INO_3$ + $NO_3^- \rightarrow IN_2O_6^-$ (Ganske et al., 2019). If $IN_2O_6^-$ is related

to $N_2O_6$, its signal increased faster than $IN_2O_5^-$ because $NO_3$ is a primary $Ce^{(IV)}$ photolysis product, then decreased faster

than $IN_2O_5^-$ because the $N_2O_6$ production rate decreased quadratically as a function of decreasing $NO_3$, whereas the $N_2O_5$

production rate remained constant following processes that converted $NO_3$ to $NO_2$. Additionally, because the aqueous phase

$NO_3+NO_3$ reaction rate is approximately 2000 times slower than that of $NO_2$ + $NO_3$ (Martin and Stevens, 1978; Katsumura

et al., 1991), even a small amount of $NO_2$ would favor the formation of $N_2O_5$ compared to $N_2O_6$. To further explore the

plausibility of $N_2O_6$ formation in this system, we conducted a theoretical investigation of the gas-phase $NO_3$ + $NO_3 \rightarrow N_2O_6$

reaction at $T$ = 298 K and $p$ = 1 atm. Quantum chemical calculations were performed using the Q-Chem 5.2 software package

(Epifanovsky et al., 2021), and molecular geometries were obtained using the B3LYP density functional (Becke, 1993) and

the 6-31G* basis set (Hariharan and Pople, 1973). All stationary points were refined by single point calculations applying the

B3LYP density functional and the cc-pVTZ basis set (Dunning, 1989) as well as CCSD(T) (Jeziorski and Monkhorst, 1981)

and the cc-pVTZ basis set. For $NO_3$ + $NO_3 \rightarrow N_2O_6$, the calculated enthalpy of reaction ($\Delta H_{rxn}$) was -35.8 kcal mol$^{-1}$ using

the CCSD(T) method, and -21.9 kcal mol$^{-1}$ using the B3LYP method. By comparison, we calculated $\Delta H_{rxn}$ values of -26.5

(CCSD(T)) and -18.1 (B3LYP) kcal mol$^{-1}$ for the $NO_3$ + $NO_2 \rightarrow N_2O_5$ reaction; the corresponding energy change ($\Delta E_{rxn}$)

values agreed within 5% of previously obtained experimental and computational $\Delta E_{rxn}$ values for this reaction (Jitariu and



Hirst, 2000; Glendening and Halpern, 2007). Thus, regardless of the quantum chemical method that was used, $NO_3 + NO_3 \rightarrow$ $N_2O_6$ appears to be an exothermic reaction, even more so than $NO_3 + NO_2 \rightarrow N_2O_5$. While the reverse reaction $N_2O_6 \rightarrow 2$

$NO_3$ is possible (although endothermic, as is $N_2O_5 \rightarrow NO_2 + NO_3$) our analysis suggests that the thermodynamically favored reaction pathway is $N_2O_6 \rightarrow N_2O_4 + O_2$, which had $\Delta H_{rxn}$ values ranging from -7.02 (CCSD(T)) to -6.15 (B3LYP) kcal $mol^{-1}$. By contrast, the reaction $N_2O_6 \rightarrow 2 NO_2 + O_2$ had $\Delta H_{rxn} = 5.28$ (CCSD(T)) - 5.58 (B3LYP) kcal $mol^{-1}$; however, because $N_2O_4 \rightarrow 2 NO_2$ is fast (Poskrebyshev et al., 2001; Atkinson et al., 2004), the overall reaction $N_2O_6 \rightarrow 2 NO_2 + O_2$ is the favored $N_2O_6$ removal pathway in the gas phase, and in solution may occur in addition to or instead of Reaction R4.

Figure 7c shows time series of $IHNO_2^-$, $HNO_2NO_3^-$, $IHNO_4^-$, and $HNO_4NO_3^-$. These ions are associated with nitrous acid ($HNO_2$) and peroxynitric acid ($HNO_4$) respectively (Veres et al., 2015). Because rapid formation of $HNO_{2-4}NO_3^-$ ions was observed following $Ce^{(IV)}$ irradiation, and because $IO_x^-$ signals were relatively low, we hypothesize that $I^- + NO_3$ charge transfer reactions were the main source of $NO_3^-$ (Lee et al., 2014), and that subsequent competitive $NO_3^- + HNO_{2-4}$ and $I^-$ $+ HNO_{2-4}$ reactions in the IMR generated both $IHNO_{2-4}^-$ and $HNO_{2-4}NO_3^-$. $HNO_4$ was generated following the reactions

$HNO_3 + hv \rightarrow OH + NO_2$, $OH + NO_3 \rightarrow HO_2 + NO_2$, and $HO_2 + NO_2 \rightarrow HNO_4$. This hypothesis is supported by the similarity between $NO_2$ and $IHNO_4^-$ time series coupled with the relatively constant concentrations of $HO_2$ generated via $OH$ $+ OH \rightarrow H_2O_2$ and $OH + H_2O_2 \rightarrow HO_2 + H_2O$ reactions. $H_2O_2$, detected as $IH_2O_2^-$, also behaved similarly as $IHO_2^-$ (Figure 7d). $HNO_2$ had a different temporal profile than the other nitrogen oxides: $IHNO_2^-$ increased throughout the experiment, and $HNO_2NO_3^-$ increased and then decreased. We hypothesize that $NO_2 + NO_2 \rightarrow N_2O_4$ and $N_2O_4 + H_2O \rightarrow HNO_2 + HNO_3$

reactions were the main source of $HNO_2$. $IN_2O_4^-$ was not detected with the CIMS following irradiation of aqueous $Ce^{(IV)}$, presumably because its hydrolysis rate was too fast (Park and Lee, 1988). In an attempt to decrease the hydrolysis rate, separate experiments were conducted in which the effluent of 40 g of irradiated solid CAN was sampled with the CIMS. At the sample sizes that were used, the solid CAN contained enough solvated $HNO_3$ and/or $H_2O$ that its irradiation provided sufficient production of nitrogen oxides for CIMS detection. As shown in Figures S8, S9 and S10, CIMS $NO_2^-$, $IN_2O_4^-$, $I(HNO_2)_n^-$, and

$(HNO_2)_nNO_3^-$ signals were significantly higher following irradiation at $\lambda = 254$ nm than at the other wavelengths, and Fig. S11 confirms that $IN_2O_4^-$ was the dominant ion signal at m/Q = 219. Taken together, these observations support our hypothesis that $HNO_2$ was generated following fast $N_2O_4$ hydrolysis in aqueous solution.

To compare measurements of reactive nitrogen and reactive oxygen species obtained from irradiated CAN/NaNO$_3$ and CAN/HNO$_3$ mixtures, Figure S12 shows time series of the same ions plotted in Figure 7 following irradiation of a solution

containing 0.5 M CAN and 3.0 M HNO$_3$ ($I_{369} \approx 7 \times 10^{15}$ photons cm$^{-2}$ s$^{-1}$). Here, 3.0 M HNO$_3$ was used because 6.0 M HNO$_3$ depleted the CIMS reagent ion too much ($IHNO_3^-:I^- \approx 15$) to achieve signal-to-noise that was sufficient for comparison to CAN/NaNO$_3$ mixtures ($IHNO_3^-:I^- \approx 3$). The same gas-phase nitrogen oxides and reactive oxygen species were observed in this reaction system as with the irradiated CAN/NaNO$_3$ mixture. The relative yields of each compound plotted in Figures 7 and S12 were within a factor of 3 of each other, although signals of nitrogen oxides and reactive oxygen species obtained from

irradiated CAN/HNO$_3$ mixtures decreased at a slower rate than the same compounds obtained from irradiated CAN/NaNO$_3$ mixtures. These trends may be due to different $Ce^{(IV)}$ composition (Fig. 3 and Sect. 3.2) and/or enhanced rate of $Ce^{(III)} +$ $NO_3 \rightarrow Ce^{(IV)}$ reactions in HNO$_3$ relative to NaNO$_3$ (Reaction R2).



### 3.6 OVOC/SOA generation from $\beta$-pinene + $NO_3$

To demonstrate proof of principle for $NO_3$-initiated oxidative aging studies, we generated $NO_3$ via irradiation of a mixture
of 0.5 M CAN and 3.0 M $HNO_3$ ($I_{369} = 7\times10^{15}$ photons $cm^{-2}$ $s^{-1}$), reacted it with $\beta$-pinene in a dark OFR, and obtained
FIGAERO-CIMS spectra of gas- and condensed-phase $\beta$-pinene + $NO_3$ oxidation products (Sect. 2.2). Figure 8a shows a spec-
trum of gas-phase $\beta$-pinene/$NO_3$ oxidation products detected between m/Q = 320 and 420, where the majority of the signal was
observed; signals shown are unmodified $(M+I)^-$ formulas. The largest ion detected was at m/Q = 356 ($IC_{10}H_{15}NO_5^-$), which
represents a major first-generation dicarbonyl nitrate oxidation product with a relative abundance of 0.31 and a calculated
saturation vapor pressure of $2\times10^{-7}$ atm ($C^* = 1900$ $\mu g$ $m^{-3}$; Claflin (2018)). Other ions corresponding to first-generation
hydroxycarbonyl nitrate ($IC_{10}H_{17}NO_5^-$, $C^* = 95$ $\mu g$ $m^{-3}$), tricarbonyl nitrate ($IC_{10}H_{15}NO_6^-$, $C^* = 35$ $\mu g$ $m^{-3}$), hydroxydicar-
bonyl nitrate ($IC_{10}H_{17}NO_6^-$, $C^* = 4.7$ $\mu g$ $m^{-3}$), and hydroxycarbonyl nitrate acid ($IC_{10}H_{17}NO_7^-$, $C^* = 0.29$ $\mu g$ $m^{-3}$) products
were detected in addition to $IC_9H_{13}NO_5^-$ and a suite of additional previously characterized $C_8$ and $C_9$ organic nitrates (Nah
et al., 2016; Takeuchi and Ng, 2019; Shen et al., 2021). The $IC_{10}H_{16}N_2O_7^-$ dinitrate was obtained following reaction of the
$\beta$-nitrooxyperoxy radical with NO or $NO_3$ (Nah et al., 2016; Bates et al., 2022). Because model-calculated NO:$NO_3$ was the
order of $10^{-5}$ under these conditions, its formation from the $RO_2$ + $NO_3$ reaction seems more likely (Orlando and Tyndall,
2012). Overall, the high molar yield and vapor pressure of $C_{10}H_{15}NO_5$ (Claflin, 2018) are consistent with it having the highest
relative abundance in the gas phase (Fig. 8a), whereas the other $C_{10}$ $\beta$-pinene oxidation products were semivolatile under our
experimental conditions.

Figure 9a shows a spectrum of condensed-phase $\beta$-pinene/$NO_3$ oxidation products obtained with the FIGAERO-CIMS;
signals were averaged over the entire thermal desorption cycle and are plotted on logarithmic scale and represent unmod-
ified $(M+I)^-$ formulas. To aid interpretation of the major features of the spectrum, bands of ion signals corresponding to
$IC_{10}H_{15}NO_x^-$, $IC_{20}H_{32}N_2O_x^-$, and $IC_{30}H_{47}N_3O_x^-$ oxidation products were highlighted and colored by the number of oxygen
atoms in their chemical formulas. Here, the largest ion detected was at m/Q = 372 ($IC_{10}H_{15}NO_6^-$), which is the condensed-
phase component of the same tricarbonyl nitrate detected in the gas-phase (Fig. 8a). $IC_{10}H_{15}NO_5^-$ and $IC_{10}H_{15}NO_{7-9}^-$ sig-
nals were also detected. The second largest ion signal was measured at m/Q = 571 ($IC_{20}H_{32}N_2O_9^-$), an acetal dimer ob-
tained from the condensed-phase reaction of two $C_{10}H_{17}NO_5$ monomers followed by $H_2O$ elimination (Claflin and Ziemann,
2018). Similar accretion reactions between other $C_{10}$ organic nitrates yielded $IC_{20}H_{32}N_2O_8^-$ and $IC_{20}H_{32}N_2O_{10-13}^-$ sig-
nals. Likewise, reactions between $C_{10}$ monomers and $C_{20}$ dimers generated $C_{30}$ trimers detected between m/Q = 768 - 864
($IC_{30}H_{47}N_3O_{12-18}^-$). The largest trimer-related ion, $IC_{30}H_{47}N_3O_{12}^-$, was generated from $C_{10}H_{17}NO_4$ + $C_{20}H_{32}NO_9$ - $H_2O$
or $C_{10}H_{17}NO_5$ + $C_{20}H_{32}NO_8$ - $H_2O$ reactions (Claflin and Ziemann, 2018). A fourth cluster of ion signals at m/Q > 984
was also observed. Unambiguous assignment of chemical formulae to these signals was challenging due to the limited range of
the CIMS $m/z$ calibration and lack of available information about $C_{>30}$ $\beta$-pinene/$NO_3$ oxidation products. However, it seems
plausible that these signals are associated with tetramers.

To compare our results with those obtained using a conventional $NO_3$ generation method (room temperature $N_2O_5$ thermal
decomposition) in an environmental chamber study, Figures 8b and 9b show reference gas- and condensed-phase FIGAERO-



I$^-$-CIMS spectra of OVOCs and SOA generated from NO$_3$ oxidation of $\beta$-pinene in the Georgia Tech environmental chamber (Takeuchi and Ng, 2019). The spectra obtained here and by Takeuchi and Ng (2019) exhibit an overall high degree of similarity, with linear correlation coefficients of 0.87 and 0.96 between the respective gas- and condensed-phase spectra. Clusters of IC$_{10}$H$_{15}$NO$_x^-$, IC$_{20}$H$_{32}$N$_2$O$_x^-$, and IC$_{30}$H$_{47}$N$_3$O$_x^-$ ion signals were present in both Figs. 9a and 9b. The main differences between the gas-phase spectra shown in Figs. 8a and 9a were the different abundances of IC$_{10}$H$_{17}$NO$_4^-$, a first-generation hydroxynitrate product (Claflin and Ziemann, 2018), and IC$_{10}$H$_{16}$N$_2$O$_7^-$. Because C$_{10}$H$_{17}$NO$_4$ is formed from RO$_2$+RO$_2$ reactions (DeVault et al., 2022) and is sufficiently volatile (C$^*$ = 750 $\mu$g m$^{-3}$) to partition into the gas phase (Claflin, 2018), differences in gas-phase C$_{10}$H$_{17}$NO$_4$ and C$_{10}$H$_{16}$N$_2$O$_7$ yields were likely related to differences in the relative rates of RO$_2$+RO$_2$ and RO$_2$ + NO$_3$ reaction pathways in the study by Takeuchi and Ng (2019) compared to this work.

## 4    Conclusions

Ce$^{(IV)}$ irradiation complements NO$_2$ + O$_3$ reactions and N$_2$O$_5$ thermal dissociation as a customizable photolytic NO$_3$ source. Important method parameters were [CAN], [HNO$_3$] or [NaNO$_3$], UV intensity, and irradiation wavelength. By contrast, important parameters for NO$_2$+O$_3$ and N$_2$O$_5$-based methods are [O$_3$], [NO$_2$], temperature, and humidity. Because Ce$^{(IV)}$ irradiation already generates NO$_3$ in aqueous solution, its performance is not hindered by humidity to the same extent (if at all) as N$_2$O$_5$-based methods, where hydrolysis of N$_2$O$_5$ to HNO$_3$ decreases the efficacy of the source. Additionally, the NO$_3$ + H$_2$O reaction rate in solution or on surfaces is slow relative to other NO$_3$ loss pathways. Another advantage of Ce$^{(IV)}$ irradiation is that it does not involve the use of O$_3$ as a reagent, therefore eliminating the possibility of competing O$_3$ and NO$_3$ oxidation of compounds that are reactive towards both oxidants (Lambe et al., 2020). To identify optimal operating conditions for maximizing [NO$_3$], we characterized concentrations of NO$_3$ at [CAN] = 10$^{-3}$ to 1 M, [HNO$_3$] = 1.0 to 6.0 M, [NaNO$_3$] = 1.0 to 4.8 M, photon flux = 6.9×10$^{14}$ to 1.0×10$^{16}$ photons cm$^{-2}$ s$^{-1}$, and irradiation wavelengths of $\lambda$ = 254, 313, 369, or 421 nm. With CAN/HNO$_3$ mixtures, maximum [NO$_3$] was achieved with [CAN] $\approx$ 0.5 M, [HNO$_3$] $\approx$ 3.0 to 6.0 M, and $I_{369}$ = 8×10$^{15}$ photons cm$^{-2}$ s$^{-1}$ (4.3 mW cm$^{-2}$). With CAN/NaNO$_3$ mixtures, maximum [NO$_3$] was achieved with [CAN] $\approx$ 1.0 M, [NaNO$_3$] $\geq$ 1.0 M, and $I_{254}$ $\approx$ 1×10$^{16}$ photons cm$^{-2}$ s$^{-1}$ (7.8 mW cm$^{-2}$). Thus, for applications such as environmental chamber or OFR studies of NO$_3$-initiated oxidative aging processes, where significant NO$_3$ production over relatively short time periods is beneficial, irradiation of concentrated Ce$^{(IV)}$ solutions at high photon flux is advantageous. Other applications that require sustained NO$_3$ production at lower concentrations and/or over longer time periods may benefit from using lower [Ce$^{(IV)}$] and photon flux. Overall, because Ce$^{(IV)}$ irradiation generates NO$_3$ at room temperature using widely-available, low-cost reagents and light sources (including high power light-emitting diodes in addition to, or instead of, UV fluorescent lamps) it is easier to apply than other NO$_3$ generation techniques - especially in field studies - and it may therefore enable more widespread studies of NO$_3$ oxidation chemistry. Adapting a photoreactor to operate with continuous injection of fresh Ce$^{(IV)}$ or alternative photolytic NO$_3$ precursors (e.g. Hering et al. (2015)) rather than in batch mode as was done here may further enhance its performance and will be investigated in future work.



*Code and data availability.* Data and KinSim mechanisms presented in this manuscript are available upon request. The KinSim kinetic solver is freely available at http://tinyurl.com/kinsim-release.

*Author contributions.* AL, BB, and PL conceived and planned the experiments. AL, BB, MA, and PL carried out the experiments. AL conceived, planned, and carried out the KinSim model simulations. NO and PZ conceived, planned, and carried out the quantum chemical calculations. AL, BB, MT, NO, PZ, MC, DW, and PL contributed to the interpretation of the results. AL took the lead in writing the manuscript. All authors provided feedback on the manuscript.

*Competing interests.* At least one of the coauthors is a member of the editorial board of Atmospheric Chemistry and Physics.

*Acknowledgements.* This work was supported by the Atmospheric Chemistry Program of the National Science Foundation: grants AGS-2131368 and AGS-2148439 to Aerodyne Research, Inc.; AGS-2131458 to the Georgia Institute of Technology; AGS-2131084 to Yale University; and AGS-2147893 to the University of Michigan. AL thanks Anita Avery, Jordan Krechmer, Timothy Onasch (Aerodyne) and Shreya Suri (Georgia Tech) for experimental assistance, Evgeni Glebov (Russian Academy of Sciences) for sharing published UV/Vis spectra of CAN/$CH_3CN$ mixtures, and the following colleagues for helpful discussions: Harald Stark, Manjula Canagaratna (Aerodyne), Steve Brown (NOAA CSL), Hartmut Herrmann (TROPOS), William Brune (Pennsylvania State University), Tyson Berg (Colorado State University), Lasse Moormann (Max Planck Institute for Chemistry), Uta Wille (University of Melbourne), Burkhard Koenig (University of Regensburg).



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



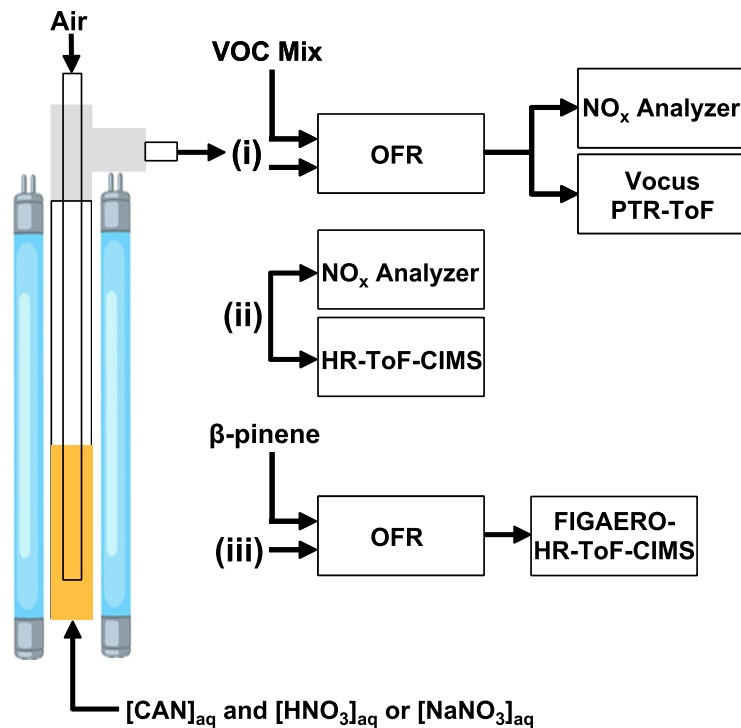

**Figure 1.** Overview of experiments conducted in this study. Aqueous mixtures of ceric ammonium nitrate (CAN) and nitric acid ($HNO_3$) or sodium nitrate ($NaNO_3$) were irradiated in a photoreactor to generate nitrate radicals ($NO_3$) in solution. Air was bubbled through the solution to evaporate $NO_3$ and other volatile photolysis products into the gas phase. The photoreactor effluent was then **(i)** injected into a dark oxidation flow reactor (OFR) along with a VOC mixture to characterize [$NO_3$] via tracer decay measurements using a Vocus proton transfer-reaction time-of-flight mass spectrometer (PTR-ToF) **(ii)** sampled with an iodide adduct high-resolution time-of-flight chemical ionization mass spectrometer (HR-ToF-CIMS) **(iii)** injected into a dark OFR to characterize $\beta$-pinene/$NO_3$ oxidation products with a Filter Inlet for Gases and Aerosols (FIGAERO) coupled to the HR-ToF-CIMS. Supporting measurements were obtained using a $NO_x$ analyzer.




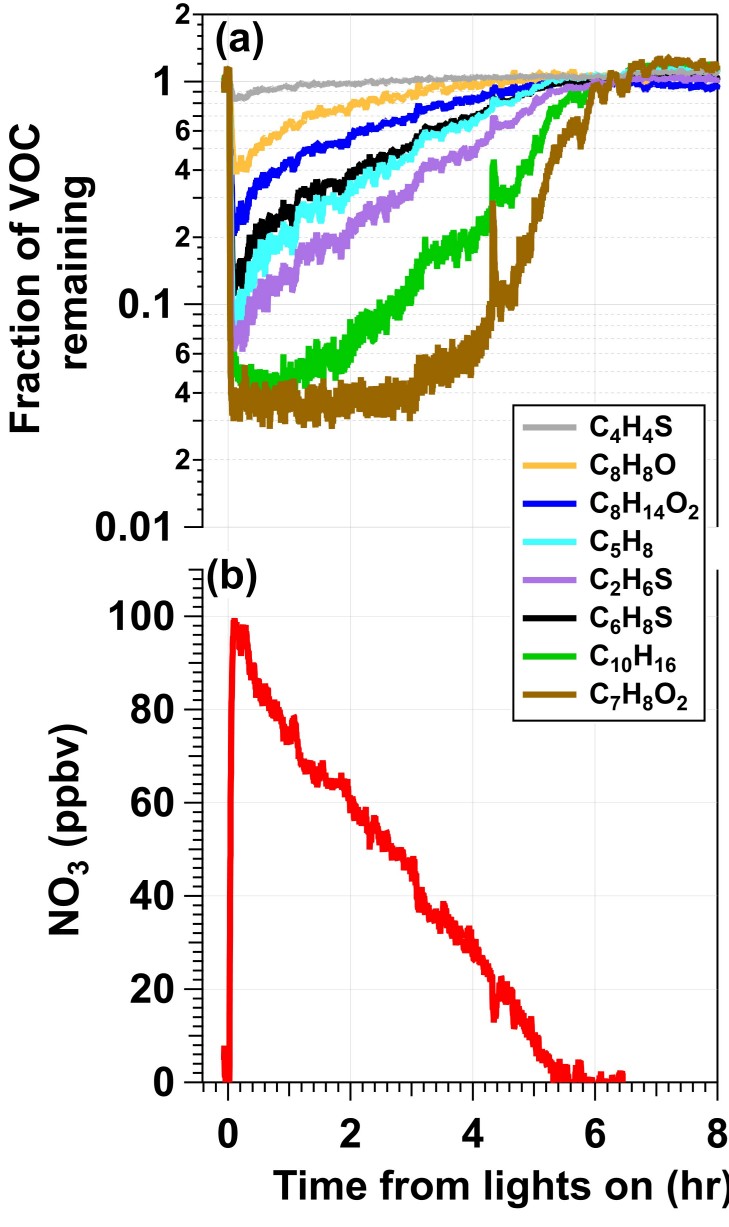

**Figure 2.** Example results from an experiment in which a mixture of 0.5 M CAN and 6.0 M HNO$_3$ was irradiated to generate NO$_3$ ($\lambda_{max}$ = 369 nm, I$_{369}$ = 7×10$^{15}$ photons cm$^{-2}$ s$^{-1}$) that was injected into the OFR along with a reactive VOC tracer mixture. **(a)** Time series of the fractional consumption of VOC tracers measured with the Vocus following irradiation: thiophene (C$_4$H$_4$S), 2,3-dihydrobenzofuran (C$_8$H$_8$O), cis-3-hexenyl-1-acetate (C$_8$H$_{14}$O$_2$), isoprene (C$_5$H$_8$), dimethyl sulfide (C$_2$H$_6$S), 2,5-dimethylthiophene (C$_6$H$_8$S), $\alpha$-pinene (C$_{10}$H$_{16}$), guaiacol (C$_7$H$_8$O$_2$). Signals of each tracer were normalized to their initial concentrations prior to NO$_3$ exposure and to acetonitrile concentrations to account for changes in the syringe pump output. **(b)** Time series of [NO$_3$] calculated from **(a)** and Tab. S2.



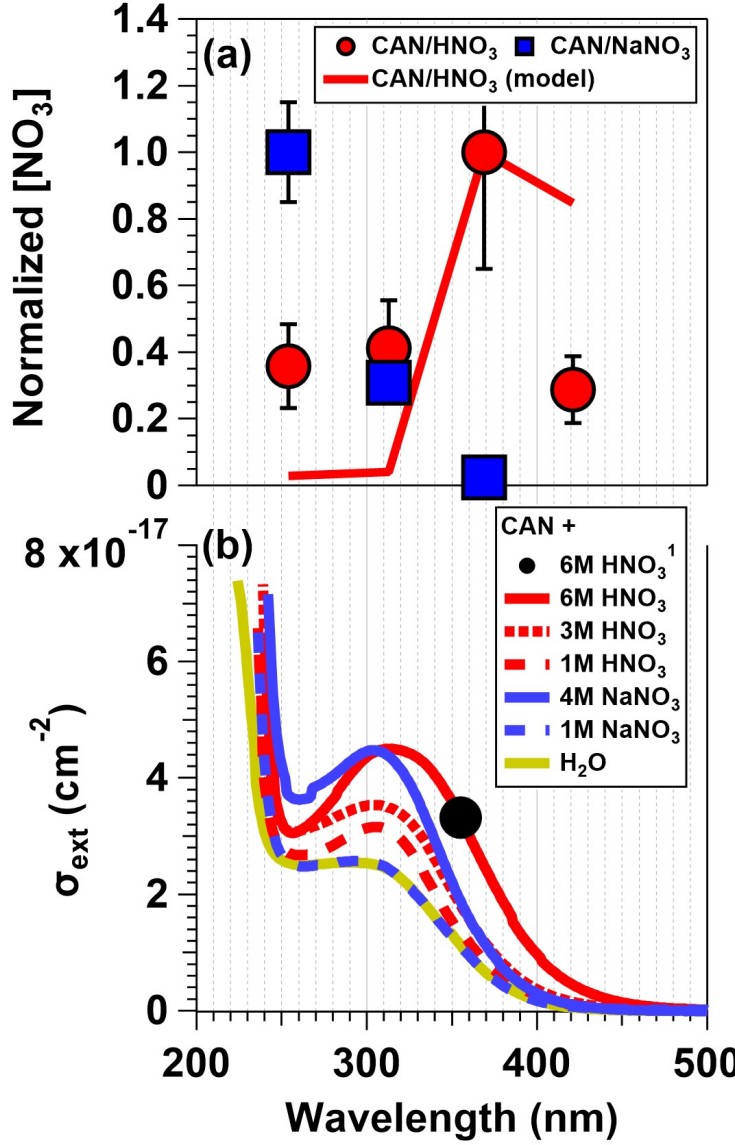

**Figure 3. (a)** [NO$_3$] values obtained from irradiated CAN & 6.0 M HNO$_3$ and CAN & 4.8 M NaNO$_3$ mixtures as a function of irradiation wavelength. Results were normalized to [NO$_3$] achieved with irradiation of CAN/HNO$_3$ mixtures at $\lambda$ = 369 nm or CAN/NaNO$_3$ mixtures at $\lambda$ = 254 nm. Error bars represent $\pm 1\sigma$ uncertainty in binned [NO$_3$] values. **(b)** Extinction cross sections ($\sigma_{ext}$) of CAN/HNO$_3$ and CAN/NaNO$_3$ mixtures (for details see Sect. 2.3). Additional figure notes: [1]: Wine et al. (1988).

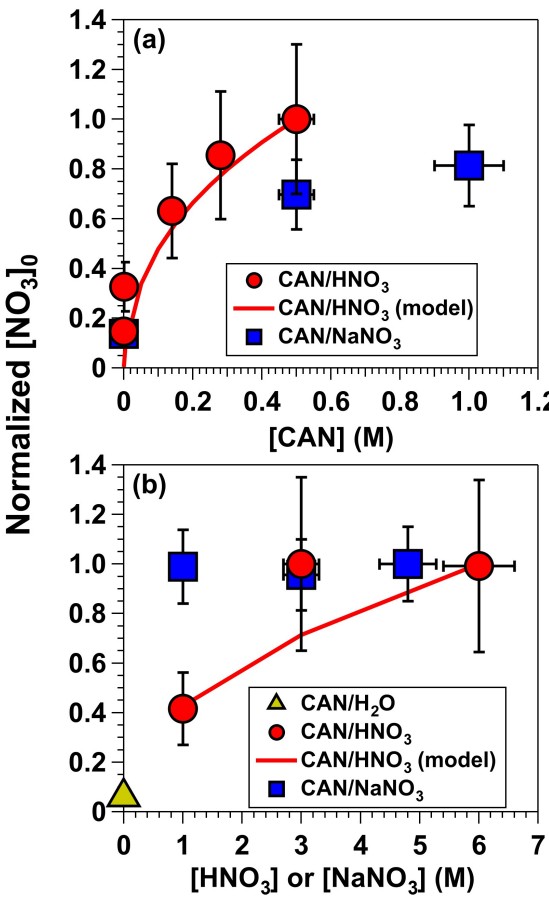

**Figure 4.** [NO$_3$] obtained from **(a)** irradiated 6.0 M HNO$_3$ solutions containing 0.001 to 0.5 M CAN ($I_{369} = 7 \times 10^{15}$ photons cm$^{-2}$ s$^{-1}$), and irradiated 1.0 M NaNO$_3$ solutions containing 0.5 to 1.0 M CAN ($I_{254} = 1 \times 10^{16}$ photons cm$^{-2}$ s$^{-1}$). **(b)** irradiated 0.5 M CAN solutions containing 1.0 to 6.0 M [HNO$_3$] or 1.0 to 4.8 M [NaNO$_3$] at the same $I_{369}$ and $I_{254}$ values used to obtain results shown in **(a)**. Results were normalized to [NO$_3$] achieved with mixtures of 0.5 M CAN and 6.0 M HNO$_3$. Error bars represent estimated ±35% uncertainty in [NO$_3$] values obtained from CAN/HNO$_3$ mixtures, ±15% uncertainty in [NO$_3$] values obtained from CAN/NaNO$_3$ mixtures, and ±10% uncertainty in [CAN], [HNO$_3$], and [NaNO$_3$] values.

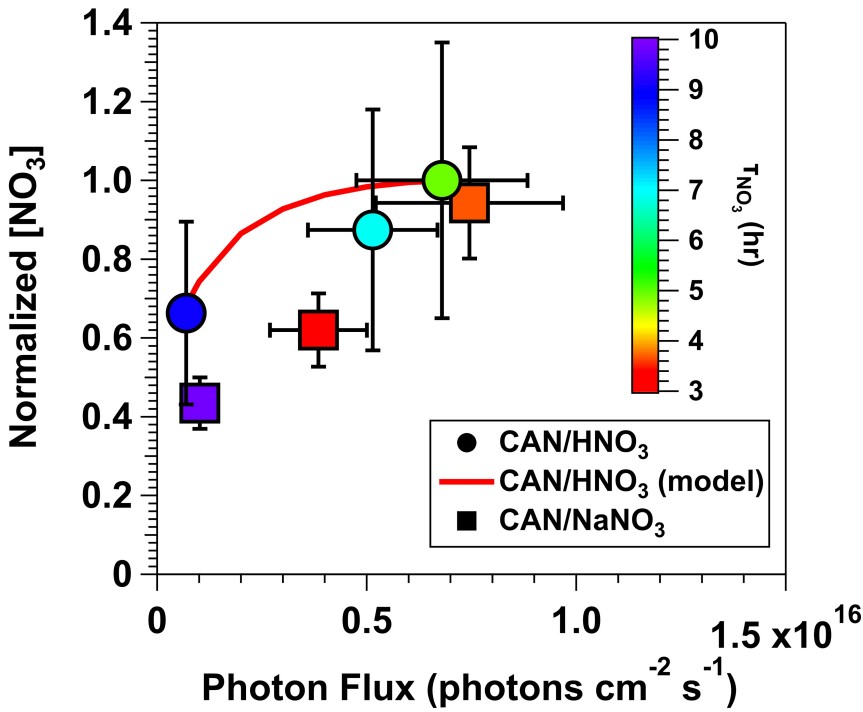

**Figure 5.** Normalized [NO$_3$] values obtained from irradiated mixtures of 0.5 M CAN and 6.0 M HNO$_3$ or 0.5 M CAN and 1.0 M NaNO$_3$ as a function of photon flux ranging from $6.9{\times}10^{14}$ to $7.5{\times}10^{15}$ photons cm$^{-2}$ s$^{-1}$. Results were normalized to [NO$_3$] achieved with 0.5 M CAN, 6.0 M HNO$_3$ and $I_{369} = 6.8{\times}10^{15}$ photons cm$^{-2}$ s$^{-1}$. Symbols are colored by the time it took for [NO$_3$] to experience one e-fold decay relative to the maximum [NO$_3$] that was measured ($\tau_{\text{NO}_3}$). Error bars represent estimated $\pm35\%$ uncertainty in [NO$_3$] values obtained from CAN/HNO$_3$ mixtures, $\pm15\%$ uncertainty in [NO$_3$] values obtained from CAN/NaNO$_3$ mixtures, and $\pm30\%$ uncertainty in photon flux values.



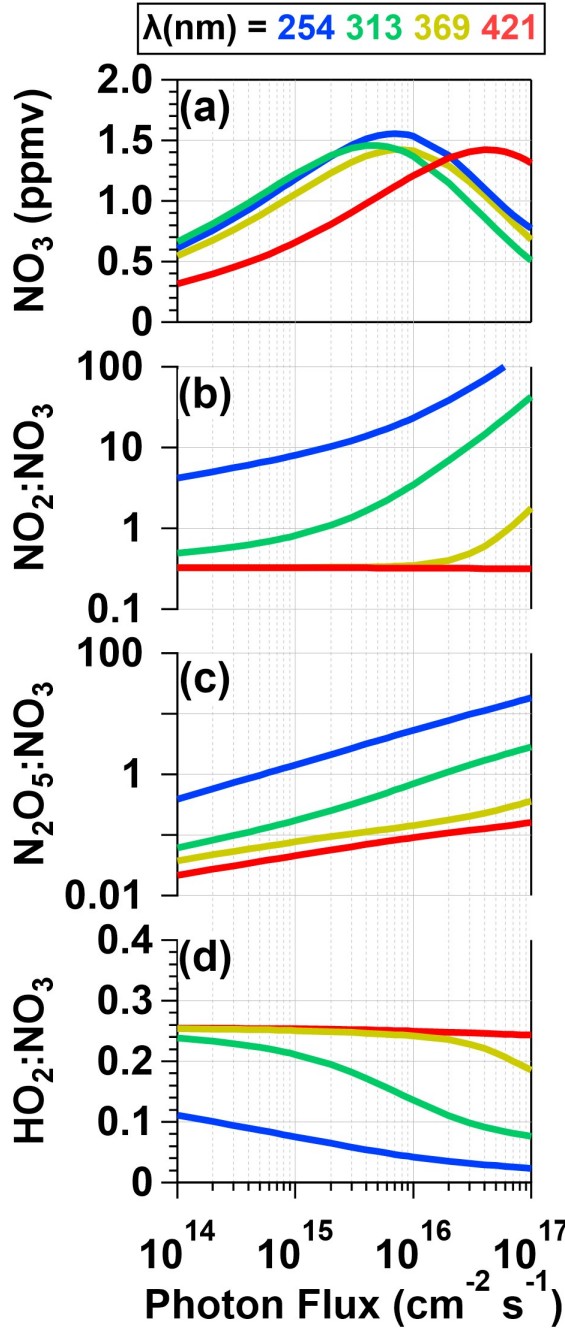

**Figure 6.** Model-calculated **(a)** $[NO_3]$, **(b)** $NO_2$:$NO_3$, **(c)** $HO_2$:$NO_3$, and **(d)** $N_2O_5$:$NO_3$ values in solution as a function of photon flux ranging from $1\times10^{14}$ to $1\times10^{17}$ photons $cm^{-2}$ $s^{-1}$ following $\lambda = 254$, 313, 369 and 421 nm irradiation of a mixture containing 0.5 M CAN and 6.0 M $HNO_3$. For details see Sect. 2.3 and Tab. S1.



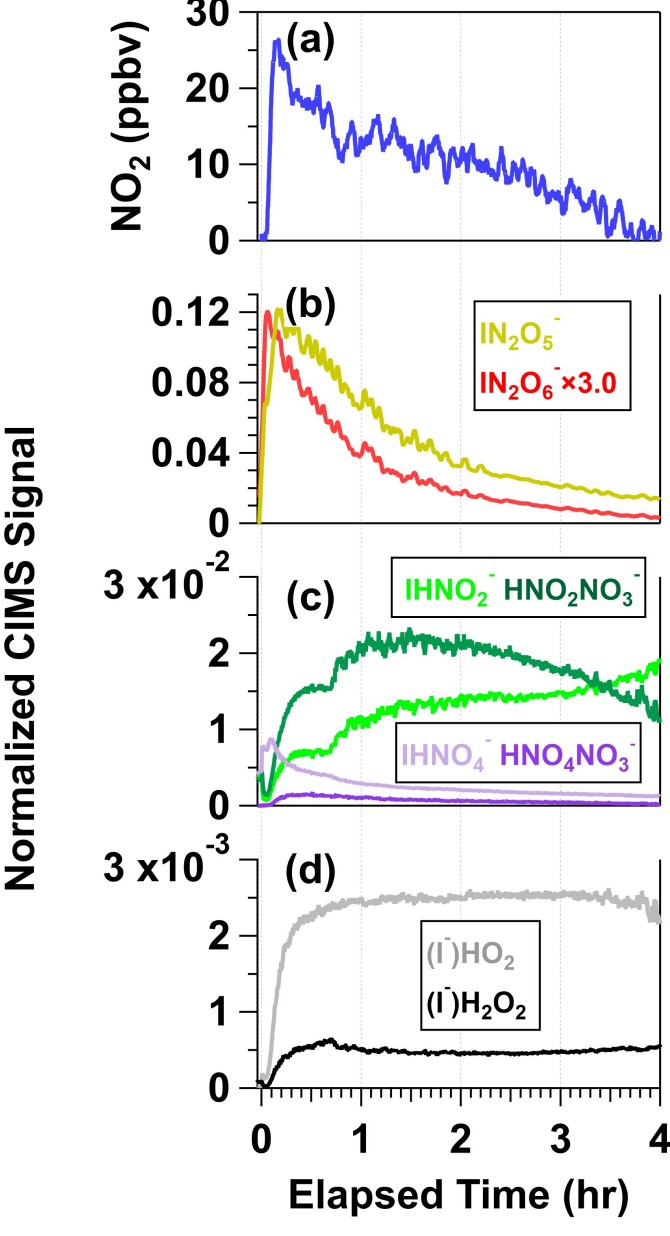

**Figure 7.** Time series of **a** $NO_2$, **(b)** $N_2O_5$ and $N_2O_6$, **(c)** $HNO_2$ and $HNO_4$, and **(d)** $HO_2$ and $H_2O_2$ detected following irradiation of a mixture containing 0.5 M CAN and 1.0 M $NaNO_3$. $N_2O_5$, $N_2O_6$, $HO_2$ and $H_2O_2$ were detected as $I^-$ adducts, and $HNO_2$ and $HNO_4$ were detected as both $I^-$ and $NO_3^-$ adducts with HR-ToF-CIMS. CIMS signals detected as iodide adducts were normalized to the $I^-$ signal prior to the start of the experiment, and CIMS signals detected as nitrate adducts were normalized to the maximum $NO_3^-$ obtained during the experiment (see Fig. S4).

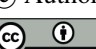



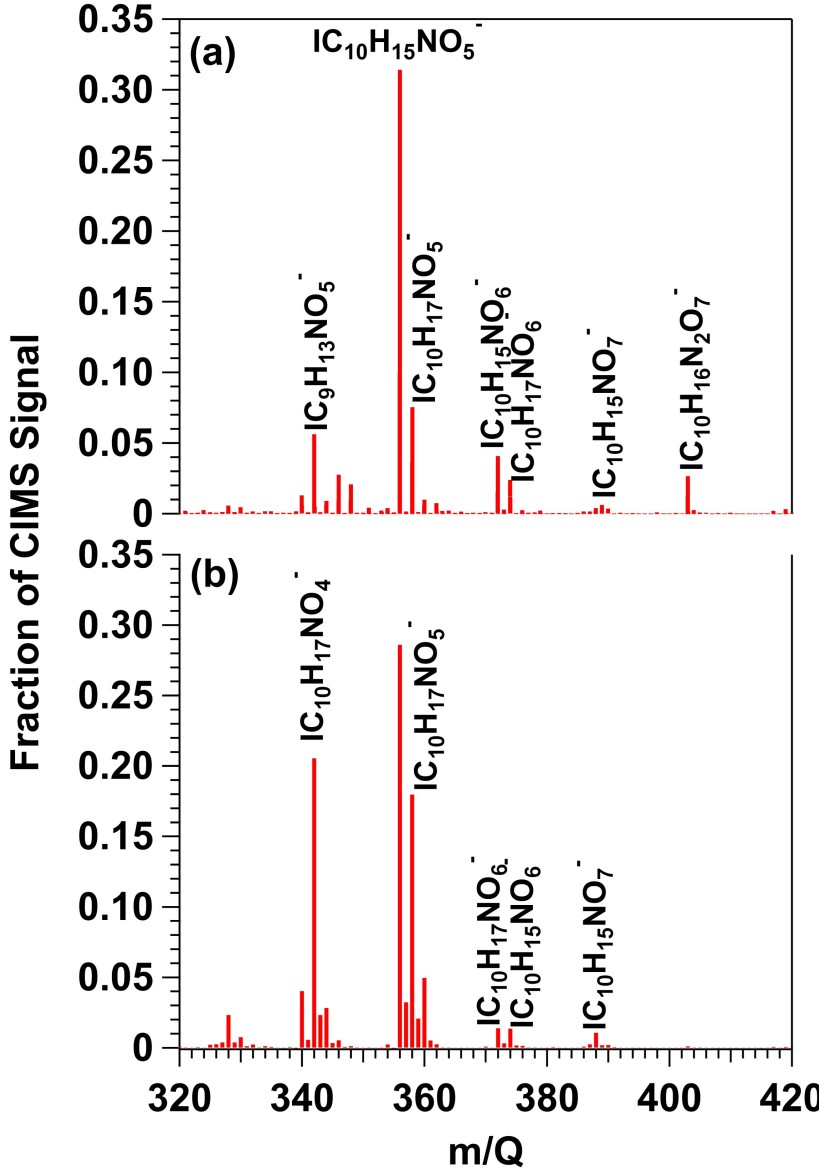

**Figure 8.** HR-ToF-CIMS spectra of gas-phase $\beta$-pinene/NO$_3$ oxidation products obtained following $\beta$-pinene reaction with NO$_3$ generated via **(a)** irradiation of a mixture of 0.5 M CAN and 3.0 M HNO$_3$ and subsequent injection into the OFR **(b)** thermal decomposition of N$_2$O$_5$ injected into the Georgia Tech environmental chamber. Signals shown are unmodified (M+I)$^-$ formulas.





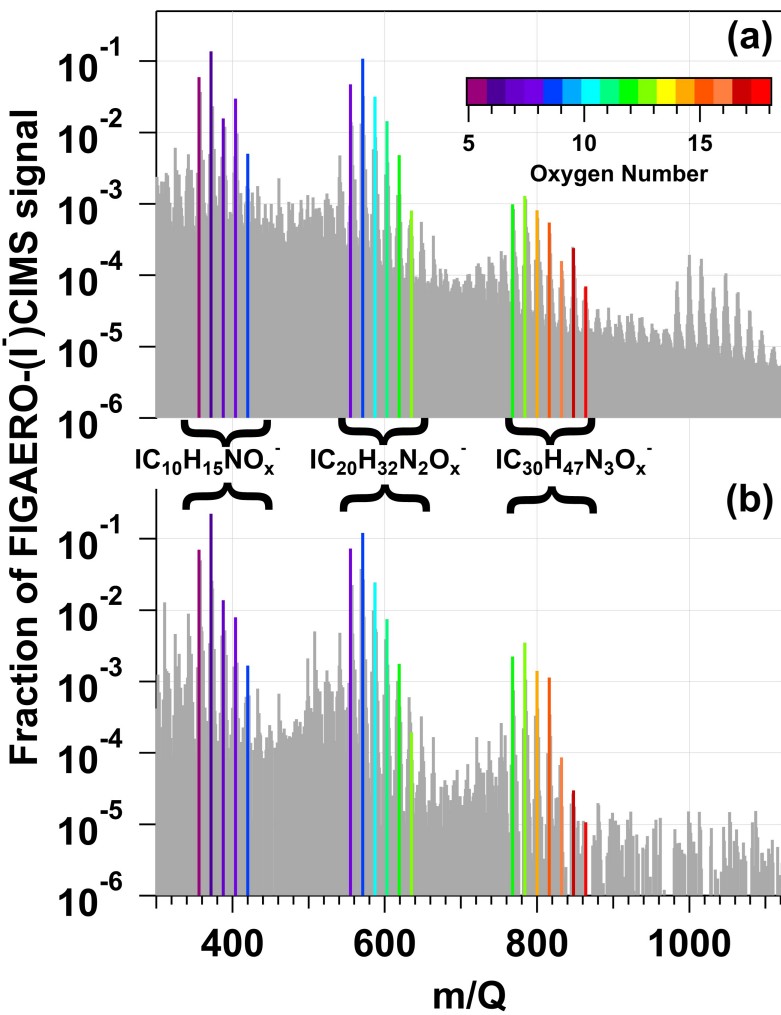

**Figure 9.** FIGAERO-HR-ToF-CIMS spectra of condensed-phase $\beta$-pinene/NO$_3$ oxidation products obtained following $\beta$-pinene reaction with NO$_3$ generated via **(a)** irradiation of a mixture of 0.5 M CAN and 3.0 M HNO$_3$ and subsequent injection into an OFR **(b)** thermal decomposition of N$_2$O$_5$ injected into the Georgia Tech environmental chamber. Signals shown are unmodified (M+I)$^-$ formulas. Bands of ion signals corresponding to C$_{10}$H$_{15}$NO$_x$, C$_{20}$H$_{32}$N$_2$O$_x$, and C$_{30}$H$_{47}$N$_3$O$_x$ oxidation products are highlighted and colored by the number of oxygen atoms in their chemical formulas.