# Peer review of "Technical note: Gas-phase nitrate radical generation via irradiation of aerated ceric ammonium nitrate mixtures"

_EGUsphere, 2023_

## Author Comment (AC1)

**Response to reviewers for the paper**
We thank the anonymous referee and Prof. Nizkorodov for their comments on our manuscript. To guide the review process, we have copied the referee's comments in black text. Our responses are in blue text. We respond to Referee #1 and #2 comments, with alterations to the paper indicated in **bold or  text** below and in annotations to the revised manuscript.

**Referee #1**

General Comments

In this manuscript the authors present a new method for creating NO3 radicals for atmospheric chemistry experiments by photolysis of aerated cerium ammonium nitrate solutions. The photolysis apparatus is described and the results of an evaluation of the effects of different experimental parameters on the modeled concentrations of species in solution and species in the gas phase based on mass spectrometer measurements is presented. The apparatus is also used to generate NO3 radicals for a reaction with beta-pinene in an oxidation flow reactor (OFR) for comparison of mass spectra of gas- and particle-phase products with the literature.

The experiments and modeling were well done, and the results are thoroughly discussed and interpreted with an excellent use of information on this solution photochemistry obtained from the literature. This method has some advantages over those currently used to create NO3 radicals and so may see considerable use by the atmospheric chemistry community. I think the manuscript is appropriate for publication after the following minor comments are addressed.

Specific Comments

1. Line 329–332: The IC10H16N2O7– ion is a hydroxy dinitrate not a dinitrate, so it could not have been formed directly from an RO2• + NO3 –> RONO2 + O2 reaction of the beta-nitrooxyperoxy radical. It would require a nitrooxyhydroxyperoxy radical, for which I am not aware of a formation pathway. Also, I did not find anywhere in the Orlando and Tyndall (2012) paper that suggests that nitrates can be formed by a RO2• + NO3 reaction, nor do any others I am aware of. The only reaction they describe is RO2• + NO3 –> RO• + NO2 + O2. I suggest you just say you don't know how this product forms.

Revised text:
L329-L332: "The I$C_{10}H_{16}N_2O_7^-$ **hydroxy** dinitrate**, which** was  **also previously observed in FIGAERO-CIMS spectra**  **of** $\alpha$-**pinene**/ NO$_3$ **SOA** (Nah et al., 2016), **was generated via an unknown reaction pathway**. ."

2. Line 367–369: Competing NO3 and O3 reactions are only a problem if one tries to synthesize NO3 radicals online. It is not an issue when one synthesizes N2O5 and then stores it in a freezer until needed. The synthesis is simple and one can easily make enough in a couple hours to last for months or years.

Revised text:

L367-369: "Another advantage of Ce(IV) irradiation is that it does not involve the use of $O_3$ as a reagent, therefore eliminating the possibility of competing $O_3$ and $NO_3$ oxidation of compounds that are reactive towards both oxidants **if $NO_2+O_3$ reactions and/or online $N_2O_5$ synthesis are used as the $NO_3$ source** (Lambe et al., 2020)."

3. Since most people using this method would be interested in knowing what RO2• reaction regime they are in, I suggest supplying a more detailed discussion of how the different synthesis conditions affect the relative concentrations of NO3, NO2, and HO2 radicals, and noting that the rate constant for RO2• + HO2 is about 10x greater than for RO2 + NO3 or NO2. It would be especially useful to say how to run the source if one wants to be in a RO2• reaction regime dominated by reactions with HO2, NO2, or NO3

4. How rapidly do gas-phase products collide with the walls in an OFR? If ROONO2 products formed from RO2• + NO2 –> ROONO2 are significant with this method then the collisions will likely be a RO2• radical sink and a source of R=O products via loss of HNO3 from ROONO2.

Because both comments consider the fate of $RO_2$, we combined our response. In a similar OFR to the one used here, Palm et al. (2016) estimated a first-order wall loss rate coefficient of 0.0025 s$^{-1}$ ($\tau_{wall}$ = 400 s) for condensable LVOC. As discussed in the revised text below, we anticipate that wall losses of alkyl $RO_2$ are negligible in the OFR because they thermally decompose within seconds. Acyl $RO_2$ species could be long-lived enough to interact with the walls, but we note that the calculated mean OFR residence time ($\tau_{OFR}$ = 120 s) is shorter than $\tau_{wall}$, thus, we anticipate that wall losses are a minor sink for acyl $RO_2$.

Revised text:

L357: "Because $C_{10}H_{17}NO_4$ is formed from $RO_2+RO_2$ […] differences in gas-phase $C_{10}H_{17}NO_4$ and $C_{10}H_{16}N_2O_7$ yields were likely related to differences in the relative rates of $RO_2+RO_2$ and $RO_2 + NO_3$ reaction pathways in the study by Takeuchi and Ng (2019) compared to this work.

**To further investigate the fate of $RO_2$ generated from VOC + $NO_3$ reactions as a function of CAN irradiation conditions, we calculated the fractional oxidative loss of generic alkyl and acyl $RO_2$ species due to reaction with $HO_2$, $NO_3$ and $NO_2$ ($F_{RO2+HO2}$, $F_{RO2+NO3}$, $F_{RO2+NO2}$) using Equations 1-3:**

$$F_{RO2+HO2} = k_{RO2+HO2}[HO_2]/(k_{RO2+HO2}[HO_2] + k_{RO2+NO3}[NO_3] + k_{RO2+NO2}[NO_2])$$
$$F_{RO2+NO2} = k_{RO2+NO3}[NO_3]/(k_{RO2+HO2}[HO_2] + k_{RO2+NO3}[NO_3] + k_{RO2+NO2}[NO_2])$$
$$F_{RO2+NO2} = k_{RO2+NO2}[NO_2]/(k_{RO2+HO2}[HO_2] + k_{RO2+NO3}[NO_3] + k_{RO2+NO2}[NO_2])$$

**Here, $k_{RO2+HO2}$, $k_{RO2+NO3}$, and $k_{RO2+NO2}$ are reaction rate coefficients for the corresponding $RO_2$ + $HO_2$, $RO_2$ + $NO_3$ and $RO_2$ + $NO_2$ forward reactions whose values are summarized in Table S3. Several simplifying assumptions were made. First, we assumed that $RO_2$ + NO reactions were negligible. Second, we did not consider $RO_2$ isomerization/autooxidation and $RO_2$ + $RO_2$ reactions that are influenced by external factors. Third, we set $F_{RO2+NO2}$ = 0 for alkyl-$RO_2$-generated $RO_2NO_2$, which thermally decompose on timescales of seconds or less (Orlando and Tyndall, 2012). Fourth, we assumed that vapor wall losses of acyl-$RO_2$-generated $RO_2NO_2$ were**

a minor RO₂ sink because the OFR residence time ($\tau_{OFR} \approx 120$ s, Sect. 2.2) was significantly shorter than their estimated wall loss timescale ($\tau_{wall} \approx 400$ s; Palm et al. (2016)). Figure 10 shows calculated $F_{RO2+HO2}$, $F_{RO2+NO3}$ and $F_{RO2+NO2}$ values for alkyl-RO₂ and acyl-RO₂ as a function of photon flux over the range of NO₃ generation conditions presented in Fig. 6. For alkyl-RO₂, $F_{RO2+HO2}$ decreased and $F_{RO2+NO3}$ increased with increasing photon flux and decreasing irradiation wavelength. On the other hand, for acyl-RO₂, $F_{RO2+NO2}$ increased while $F_{RO2+HO2}$ and $F_{RO2+NO3}$ decreased over the same irradiation conditions. Overall, at the optimal NO₃ generation conditions (e.g. $\lambda = 369$ nm and $I_{369} \approx 10^{16}$ photons cm⁻² s⁻¹), our calculations suggest that $F_{RO2+HO2} \approx F_{RO2+NO3}$ for alkyl-RO₂ (Figs. 10c) and that $F_{RO2+HO2} \approx F_{RO2+NO3} \approx F_{RO2+NO2}$ for acyl-RO₂ (Fig. 10g).

We added the following figure to the revised manuscript:

[Figure]

We added the following citation to the revised manuscript:

B.B. Palm, P. Campuzano-Jost, A.M. Ortega, D.A. Day, L. Kaser, W. Jud, T. Karl, A. Hansel, J.F. Hunter, E.S. Cross, J.H. Kroll, A. Turnipseed, Z. Peng, W.H. Brune, and J.L. Jimenez. In situ secondary organic aerosol formation from ambient pine forest air using an oxidation flow reactor. *Atmospheric Chemistry and Physics*, Atmos. Chem. Phys., 16, 2943-2970, doi:10.5194/acp-16-2943-2016, 2016.

We added the following table to the revised supplement:

**Table S3.** Room-temperature bimolecular rate coefficients ($k_{298}$) used to calculate fates of alkyl and acyl organic peroxy radicals (alkyl-$RO_2$, acyl-$RO_2$) formed from VOC + $NO_3$ reactions as a function of CAN irradiation conditions summarized in Fig. 6. Kinetic data is adapted from Orlando and Tyndall (2012). Rate coefficients are given in units of $cm^3$ $molecule^{-1}$ $s^{-1}$.

| Reactant 1 | Reactant 2 | $k_{298}$ |
|---|---|---|
| alkyl-$RO_2$ | $HO_2$ | $7.7 \times 10^{-12}$ |
| acyl-$RO_2$ | $HO_2$ | $1.4 \times 10^{-11}$ |
| alkyl-$RO_2$ | $NO_3$ | $2.4 \times 10^{-12}$ |
| acyl-$RO_2$ | $NO_3$ | $3.2 \times 10^{-12}$ |
| acyl-$RO_2$ | $NO_2$ | $1.1 \times 10^{-11}$ |

**Referee #2**

I reviewed this paper to fill in for missing second review to avoid further delays in the open discussion process.

This is well written manuscript that proposes a new way of producing flows containing NO3 for atmospheric experiments. The system is based on (complex) photochemistry of Ce(IV) nitrate, and the bulk of the manuscript describes tests in which concentrations, irradiation wavelengths, and light fluxes are varied to find the optimum setting for making NO3. The system is then tested by making SOA from NO3 produced by Ce(IV) nitrate and by conventional N2O5 thermal decomposition. I only have minor comments

CONTENT

1. I think section 3.5, especially the discussion of possible N2O6 formation, may distract the readers from the main message of the manuscript. This discussion is more pertinent to ionization chemistry in I- CIMS than to the topic of characterizing the NO3 source. I would suggest shortening this section (or maybe even removing it and developing it into a stand-alone paper). But keeping it there is OK also, as I- CIMS if fairly common, and the discussion will be user to I- CIMS users who work with this NO3 source.

In our opinion some discussion of non-$NO_3$ photolysis products is useful because if nothing else it may help readers to improve experimental design, especially with regards to the presence/formation of $HNO_2$, $HNO_3$, $HNO_4$, and/or $H_2O_2$. With that said, we are receptive to your suggestion to shorten this section. We shortened the text in this section by ~50% by removing some text completely and moving the following text to the Supplement.

**Section S1. Theoretical analysis of the gas-phase $NO_3$ + $NO_3$ → $N_2O_6$ reaction**

We conducted a theoretical investigation of the gas-phase $NO_3$ + $NO_3$ → $N_2O_6$ reaction at T = 298 K and p = 1 atm. Quantum chemical calculations were performed using the Q-Chem 5.2 software package (Epifanovsky et al., 2021), and molecular geometries were obtained using the B3LYP density functional (Becke, 1993) and the 6-31G* basis set (Hariharan and Pople, 1973). All stationary points were refined by single point calculations applying the B3LYP density functional and the cc-pVTZ basis set (Dunning, 1989) as well as CCSD(T) (Jeziorski and Monkhorst, 1981) and the cc-pVTZ basis set. For $NO_3$ + $NO_3$ → $N_2O_6$, the calculated enthalpy of reaction ($\Delta H_{rxn}$) was -35.8 kcal $mol^{-1}$ using the CCSD(T) method, and -21.9 kcal $mol^{-1}$ using the B3LYP method. By comparison, we calculated $\Delta H_{rxn}$ values of -26.5 (CCSD(T)) and -18.1 (B3LYP)

kcal mol$^{-1}$ for the NO$_3$ + NO$_2$ → N$_2$O$_5$ reaction; the corresponding energy change ($\Delta E_{rxn}$) values agreed within 5% of previously obtained experimental and computational $\Delta E_{rxn}$ values for this reaction (Jitariu and Hirst, 2000; Glendening and Halpern, 2007). Thus, regardless of the quantum chemical method that was used, NO$_3$ + NO$_3$ → N$_2$O$_6$ appears to be an exothermic reaction, even more so than NO$_3$ + NO$_2$ → N$_2$O$_5$. While the reverse reaction N$_2$O$_6$ → 2 NO$_3$ is possible (although endothermic, as is N$_2$O$_5$→ NO$_2$ + NO$_3$) our analysis suggests that the thermodynamically favored reaction pathway is N$_2$O$_6$ → N$_2$O$_4$ + O$_2$, which had $\Delta H_{rxn}$ values ranging from -7.02 (CCSD(T)) to -6.15 (B3LYP) kcal mol$^{-1}$. By contrast, the reaction N$_2$O$_6$ → 2 NO$_2$ + O$_2$ had $\Delta H_{rxn}$ = 5.28 (CCSD(T)) and 5.58 (B3LYP) kcal mol$^{-1}$ ; however, because N$_2$O$_4$ → 2 NO$_2$ is fast (Poskrebyshev et al., 2001; Atkinson et al., 2004), the overall reaction N$_2$O$_6$ → 2 NO$_2$ + O$_2$ is the favored N$_2$O$_6$ removal pathway in the gas phase, and in solution may occur in addition to or instead of Reaction R4.

**Section S2 Discussion of additional I$^-$ CIMS signals**

**Section S2.1 IO$_x^-$, NO$_2^-$, NO$_3^-$, IHNO$_3^-$, HNO$_3$NO$_3^-$, and IN$_2$O$_7^-$**

Figure S4 shows time series of I$^-$, IH$_2$O$^-$, IO$^-$, IO$_2^-$, NO$_2^-$, NO$_3^-$, IHNO$_3^-$, and HNO$_3$NO$_3^-$ obtained with the CIMS following irradiation of a mixture of 0.5 M CAN and 1.0 M NaNO$_3$. Signals of I$^-$, IH$_2$O$^-$ and IHNO$_3^-$ decreased following irradiation of the CAN/NaNO$_3$ mixture, whereas IO$^-$, IO$_2^-$, NO$_2^-$, NO$_3^-$, and HNO$_3$NO$_3^-$ One potential source of IO$_x^-$ is I$^-$ + O$_3$ reactions in the CIMS IMR; if this reaction was the sole source of IO$_x^-$ here, we estimate an upper limit O$_3$ mixing ratio of approximately 15 ppbv present in the IMR (Dörich et al., 2021). NO$_2^-$ is generated following the reaction of I$^-$ and/or IO$_x^-$ with HNO$_2$ (Abida and Osthoff, 2011), and NO$_3^-$ is generated from the reaction of I$^-$ and/or IO$_x^-$ with multiple nitrogen oxides, including NO$_3$, HNO$_3$, HNO$_4$, and N$_2$O$_5$ (Huey et al., 1995; Veres et al., 2015; Dörich et al., 2021). Figure S4 additionally shows a time series of IN$_2$O$_7^-$, and Figures S5, S6, and S7 shows high-resolution CIMS spectra at m/Q = 235, 251, and 267.  Given IN$_2$O$_7^-$:IN$_2$O$_5^-$ ≈ 10$^{-3}$ coupled with similar IN$_2$O$_5^-$ and IN$_2$O$_7^-$ temporal profiles (Fig. S4), we hypothesize that N$_2$O$_5$+ IO$_x^-$ reactions in the IMR were the primary source of IN$_2$O$_7^-$.

**Section S2.2 IN$_2$O$_4^-$**

IN$_2$O$_4^-$ was not detected with the CIMS following irradiation of aqueous Ce(IV), presumably because its hydrolysis rate was too fast (Park and Lee, 1988). In an attempt to decrease the hydrolysis rate, separate experiments were conducted in which the effluent of 40 g of irradiated solid CAN was sampled with the CIMS. At the sample sizes that were used, the solid CAN contained enough solvated HNO$_3$ and/or H$_2$O that its irradiation provided sufficient production of nitrogen oxides for CIMS detection. As shown in Figures S8, S9 and S10, CIMS NO$_2^-$, IN$_2$O$_4^-$, I(HNO$_2$)$_n^-$, and (HNO$_2$)$_n$NO$_3^-$ signals were significantly higher following irradiation at λ = 254 nm than at the other wavelengths, and Fig. S11 confirms that IN$_2$O$_4^-$ was the dominant ion signal at m/Q = 219. Taken together, these observations support our hypothesis that HNO$_2$ was generated following fast N$_2$O$_4$ hydrolysis in aqueous solution.

EDITORIAL

2. L82: I would mention that flux at 421 nm was not quantified

Revised text:
L82: "To quantify the photon flux *I* in the photoreactor for studies that used λ = 254, 313, or 369 nm radiation, we measured the rate of externally added O$_3$ (λ = 254 nm) or NO$_2$ photolysis (λ = 313 or 369

nm) as a function of lamp voltage under dry conditions (RH < 5%). **The photon flux was not quantified in studies that used λ = 421 nm radiation.**"

3. L85, L170, etc.: I would suggest replacing "I-values" with a more explicit name

Revised text:

L82-L85: "To quantify the photon flux $I_\lambda$ in the photoreactor for studies that used λ = 254, 313, or 369 nm radiation, […] formation. ⊢**Photon flux** values were then calculated…"

L169-L170: "These differences in [$NO_3$] were larger than the differences in calibrated ⊢ **photon flux** values at the maximum output of each lamp type (±40%; Sect. 2.1)."

L231: "[$NO_3$] values decreased at higher ⊢ **photon flux** values due to conversion of $NO_3$ to $NO_2$ via photolysis."

4. L94: Table S2 is referred to before Table S1 is mentioned on line 118. Probably best to fix the order.

Thank you. We have switched the order of Tables S1 and S2 in the supplement and the corresponding references in the text.

5. L102: do you have an estimate of the effect of RO2 reactions, perhaps from OFR modelling?

Thank you for your question and suggestion. To investigate this, we constructed a simple kinetic model to calculate the mean $RO_2$ concentration generated from $NO_3$ oxidation of the VOC tracers in the OFR over 120 sec residence time. We assumed $NO_3$ oxidation of each VOC generated the same generic/lumped $RO_2$ and that this $RO_2$ reacted with $NO_3$ at a rate coefficient of $2.4*10^{-12}$ $cm^3$ $molecule^{-1}$ $s^{-1}$ (Table S3, see response to Comment #4 by Referee #1). The model was initialized using the VOC concentrations listed in Table S2 (now Table S1) and the initial $NO_3$ mixing ratio was varied between 1 and 1000 ppbv. We then calculated the external $NO_3$ reactivity of this lumped $RO_2$ species and normalized it to the total external $NO_3$ reactivity of the VOC tracers (5 $s^{-1}$). The result is shown below:

[Figure]

*Figure S2.* Fractional NO3 consumption by RO2 generated from VOC + NO3 reactions during NO3 characterization studies described in Sect. 3.1.

We interpret this result as follows: at the lowest $[NO_3]$, less $RO_2$ is generated because less VOC is consumed by $NO_3$, whereas at the highest $[NO_3]$, the VOC tracers (and $RO_2$) were consumed quickly enough that sustaining high $RO_2$ concentrations was more difficult. At an initial $NO_3$ mixing ratio of 50 ppbv, a maximum of ~17% of the $NO_3$ was consumed by $RO_2$.

Revised text:

L102: "Here, we assumed that the total concentration of reacted VOCs was equal to the concentration of $NO_3$ injected into the OFR. **B**ecause $NO_3$ may additionally react with organic peroxy radicals ($RO_2$) generated from VOC + $NO_3$ reactions as well as OVOCs, these calculated $NO_3$ concentrations represent lower limits. **Modeling calculations suggest that the fractional consumption of $NO_3$ by $RO_2$ ranged from <0.01 to 0.17 over the range of conditions that were studied (Fig. S2)**."

Figure S2 was added to the revised supplement.

6.  Figure 3: instead of saying "Additional figure notes" you can say "The black dot corresponds to data from Wine et al. (1988)"

Revised Figure 3 caption:

"Figure 3. (a) $[NO_3]$ values obtained from irradiated CAN & 6.0 M $HNO_3$ and CAN & 4.8 M $NaNO_3$ mixtures as a function of irradiation wavelength. Results were normalized to $[NO_3]$ achieved with irradiation of CAN/$HNO_3$ mixtures at λ = 369 nm or CAN/$NaNO_3$ mixtures at λ = 254 nm. Error bars represent ±1σ uncertainty in binned $[NO_3]$ values. (b) Extinction cross sections ($\sigma_{ext}$) of CAN/$HNO_3$ and CAN/$NaNO_3$ mixtures (for details see Sect. 2.3).  **The black dot corresponds to data from Wine et al. (1988)**."

7.  L216 and Figure 5: I presume the data are for 369 nm only. I would mention it here and in the figure caption.

The data for CAN/$HNO_3$ mixtures are for 369 nm only, but the data for CAN/$NaNO_3$ mixtures are for 254 nm. Revised text:

L216: "Figure 5 shows normalized [NO₃] values obtained from  irradiated mixtures of 0.5 M CAN & 6.0 M HNO₃ **(λ = 369 nm)** and  0.5 M CAN & 1.0 M NaNO₃ **(λ = 254 nm)** as a function of photon flux ranging from $6.9\times10^{14}$ to $7.5\times10^{15}$ photons cm$^{-2}$ s $^{-1}$ . Results **for both CAN/HNO₃ and CAN/NaNO₃ mixtures** were normalized to [NO₃] achieved with 0.5 M CAN, 6.0 M HNO₃ and $I_{369}$ = $6.8\times10^{15}$ photons cm$^{-2}$ s$^{-1}$"

Figure 5 caption: "Normalized [NO₃] values obtained from irradiated mixtures of 0.5 M CAN and 6.0 M HNO₃ **(λ = 369 nm)** or 0.5 M CAN and 1.0 M NaNO₃ **(λ = 254 nm)** as a function of photon flux ranging from $6.9\times10^{14}$ to $7.5\times10^{15}$ photons cm$^2$ s $^{-1}$."

8. Figure 7 caption: a NO2 -> (a) NO2

Thank you – the correction has been made.

9. Figure 7: would it make sense to also include NO3 mixing ratio in this figure measured under the same conditions, similar to the one in Figure 2?

Thank you for your suggestion. A revised version of Figure 7 is shown below:

[Figure]

Revised Figure 7 caption: "Figure 7. Time series of (a) NO₂ **and NO₃** (b) N₂O₅ and N₂O₆ …"

The pertinent revised text in the first paragraph of the (revised) Section 3.5:

"Figure 7 shows time series of reactive nitrogen and reactive oxygen species detected following irradiation of a mixture of 0.5 M CAN and 1.0 M NaNO$_3$ ($I_{254} \approx 10^{16}$ photons cm$^{-2}$ s$^{-1}$ ) shown here because the signal-to-noise in CIMS measurements of irradiated CAN/NaNO$_3$ mixtures was generally better than in measurements of irradiated CAN/HNO$_3$ mixtures due to reagent ion depletion by HNO$_3$. **A time series of [NO$_3$] obtained from VOC tracer decay measurements in a separate experiment under similar irradiation conditions is also shown.** The NO$_2$ **and NO$_3$** mixing ratio**s** reached maximum values of 26 **and 58** ppbv shortly after the lights were turned on (Fig. 7a), suggesting an initial NO$_2$:NO$_3$ ≈ **0.45**."

Although it was not requested by the reviewer here, to facilitate comparison of measured and modeled NO$_3$ values in Figure 6, we also calculated the mean NO$_3$ mixing ratio over the first 4 hours of the experiment shown in Fig. 2 (59.3 ppbv) (to correspond to the 4 hour model simulation time (L132)) used to generate the results that are shown in Fig. 6). The revised Figure 6 is shown below:

[Figure]

Revised Figure 6 caption:

Figure 6. Model-calculated (a) [NO$_3$], (b) NO$_2$:NO$_3$, (c) HO$_2$:NO$_3$, and (d) N$_2$O$_5$:NO$_3$ values in solution as a function of photon flux ranging from 1×10$^{14}$ to 1×10$^{17}$ photons cm$^{-2}$ s$^{-1}$ following λ = 254, 313, 369 and 421 nm irradiation of a mixture containing 0.5 M CAN and 6.0 M HNO$_3$. **[NO$_3$] obtained from measurements shown in Fig. 2 is plotted in (a)**. For details see Sect. 2.3 and Tab. S1.

Revised text:

L226: "To examine concentrations of NO$_3$ and a subset of additional gas-phase photolysis products obtained over a wider range of conditions, Figure 6 plots model-calculated [NO$_3$], NO$_2$:NO$_3$, HO$_2$:NO$_3$, and N$_2$O$_5$:NO$_3$ values as a function of photon flux ranging from 1×10$^{14}$ to 1×10$^{17}$ photons cm$^{-2}$ s$^{-1}$ following λ = 254, 313, 369 and 421 nm irradiation of a mixture of 0.5 M CAN and 6.0 M HNO$_3$. **Figure 6a also plots the measured [NO$_3$] obtained from irradiation of a mixture of 0.5 M CAN and 6.0 M HNO$_3$ at $I_{369}$ = 7×10$^{15}$ photons cm$^{-2}$ s$^{-1}$ (Fig. 2) after correcting for dilution between the photoreactor and the OFR (Sect. 2.2)**

**and application of a NO$_3$ wall loss rate coefficient of 0.2 s$^{-1}$ within the photoreactor (Dubé et al., 2006).** **At this photon flux value, the model-calculated [NO$_3$] = 1.4 ppmv agrees with [NO$_3$] = 1.7 ± 0.6 ppmv obtained from measurements.** When considering only the primary photochemical process (Reactions R1-R5), maximum [NO$_3$] values within ± 10% of each other were achieved at photon fluxes ranging from 5×10$^{15}$ (λ = 313 nm) to 4×10$^{16}$ photons cm$^{-2}$ s$^{-1}$ (λ = 421 nm).

The following citation was added to References:

Dubé, W. P., Brown, S. S., Osthoff, H. D., Nunley, M. R., Ciciora, S. J., Paris, M. W., McLaughlin, R. J., and Ravishankara, A. R.: Aircraft instrument for simultaneous, in situ measurement of NO3 and N2O5 via pulsed cavity ring-down spectroscopy, Rev. Sci., 77, 34–101, https://doi.org/10.1063/1.2176058, 2006.

    10. L330: was the order -> was of the order

Revised text: L330: "Because model-calculated NO:NO$_3$ was **on** the order of 10$^{-5}$ under these conditions…"

    11. Figure S2: captions mentions acetonitrile but it is not clear what role in plays in the data as normalization is done with respect to thiophene. Was everything first normalized to acetonitrile as the text states? I would mention this in the caption more explicitly.

That is correct – to clarify, we paraphrased the text from L145-L147 in the revised Figure S2 caption.

Revised text:

Figure S2. "Relative rate coefficients obtained from Vocus measurements of acetonitrile **(C$_2$H$_3$N)**, thiophene (C$_4$H$_4$S), 2,3-dibenzofuran (C$_8$H$_8$O), and cis-3-hexynyl-acetate (C$_8$H$_{14}$O$_2$) tracers used in characterization studies described in Sect. 3.2. **Here, concentrations of C$_4$H$_4$S, C$_8$H$_8$O, and C$_8$H$_{14}$O$_2$ were first normalized to the C$_2$H$_3$N concentration to correct for changes in the syringe pump output over time and then normalized to the VOC concentration prior to NO$_3$ exposure**. Literature relative rate coefficients obtained from kinetic data published by Atkinson (1991) and Atkinson et al. (1995)."

    12. Table S1: It appears the authors list absorption cross section (or perhaps a product of that with the quantum yield) instead of the photolysis rate constants for photolysis processes. I would note this to avoid confusion.

We fixed a typo in the Table S1 caption by changing the units of the noted absorption cross sections (cm$^{-2}$ to cm$^2$) and added some clarification text:

Table S1: "KinSim mechanism used to calculate concentrations of species associated with irradiation of CAN/HNO$_3$ mixtures. Rate coefficients **(blue text, red text or black text)** or absorption cross sections (**teal text**) are given in units of cm$^3$ molecules$^{-1}$ s$^{-1}$ (blue text), M$^{-1}$ s$^{-1}$(red text), cm$^2$ (teal text), or s$^{-1}$ (black text)."

    13. Table S1: when multiple values for the same process are listed, such as the Martin and Stevens (1978) values for Ce(III) + NO3 reaction, which one is being used in the model?

For results presented in the main paper, we used values corresponding to the 6.0 M $HNO_3$ case. The other values correspond to 1.0 M and 3.0 M $HNO_3$ solutions, indicated by subscripts defined in the last line of the Table S1 caption: "Additional table notes: [1][$HNO_3$]=1.0 M, [2][$HNO_3$]=3.0 M, [3][$HNO_3$]=6.0 M" and referenced in the "Citation" column of the table. This notation was chosen mainly for brevity to allow the table to fit on the page. To make the notation more intuitive, we additionally color-coded the applicable values in the "Citation" column of this table.

Revised text in the last sentence of table caption:

"**Citations of rate coefficient and absorption cross section values that are specifically applicable to mixtures containing [$HNO_3$] = 1.0 M, 3.0 M or 6.0 M are colored with brown, violet, or orange text, respectively**."

Revised table:

| Reactant 1 | Reactant 2 | Product 1 | Product 2 | Product 3 | RateCoeff | Citation |
|---|---|---|---|---|---|---|
| $Ce^{IV}NO_3^-$ | $h\nu_{254}$ | $[Ce^{III}...NO_3]^*$ | $NO_3$ | | 2.7E-17 | this work |
| | | | | | 3.1E-17 | this work |
| | | | | | 3.1E-17 | this work |
| $Ce^{IV}NO_3^-$ | $h\nu_{313}$ | $[Ce^{III}...NO_3]^*$ | $NO_3$ | | 3.1E-17 | this work |
| | | | | | 3.5E-17 | this work |
| | | | | | 4.5E-17 | this work |
| $Ce^{IV}NO_3^-$ | $h\nu_{369}$ | $[Ce^{III}...NO_3]^*$ | $NO_3$ | | 8.7E-18 | this work |
| | | | | | 1.2E-17 | this work |
| | | | | | 2.5E-17 | this work |
| $Ce^{IV}NO_3^-$ | $h\nu_{421}$ | $[Ce^{III}...NO_3]^*$ | $NO_3$ | | 1.0E-18 | this work |
| | | | | | 1.5E-18 | this work |
| | | | | | 4.4E-18 | this work |
| $[Ce^{III}...NO_3]^*$ | | $Ce^{IV}NO_3^-$ | | | 5.12E+04 | Martin and Stevens (1978) |
| | | | | | 6.30E+03 | Martin and Stevens (1978) |
| | | | | | 0 | Martin and Stevens (1978) |
| $[Ce^{III}...NO_3]^*$ | | $Ce^{III}$ | $NO_3$ | | 4.36E+04 | Martin and Stevens (1978) |
| | | | | | 6.76E+04 | Martin and Stevens (1978) |
| | | | | | 7.74E+04 | Martin and Stevens (1978) |
| $Ce^{III}$ | $NO_3$ | $Ce^{IV}NO_3^-$ | | | 6.00E+07 (9.96E-14) | Martin and Stevens (1978) |
| | | | | | 1.08E+06 (1.79E-15) | Martin and Stevens (1978) |
| | | | | | 1.78E+06 (2.96E-15) | Martin and Stevens (1978) |